# LearnIR: Learnable Posterior Sampling for Real-World Image Restoration

**Yihang Bao**[1*]   **Zhen Huang**[2*]   **Shanyan Guan**[2]   **Songlin Yang**[2]
**Yanhao Ge**[2]   **Wei Li**[2]   **Bukun Huang**[3]   **Zengmin Xu**[1,4,5,†]

[1] School of Mathematics and Computing Science, Guangxi Colleges and Universities Key Laboratory of Data Analysis and Computation, Guilin University of Electronic Technology,
[2] vivo Mobile Communication Co., Ltd
[3] Zhejiang Gongshang University
[4] Center for Applied Mathematics of Guangxi (GUET)
[5] Anview.ai

`byh11@mails.guet.edu.cn`   `xzm@guet.edu.cn`
`{huangzhen.hzaiyj, guanshanyan, songlin.yang, halege, liwei.yxgh}@vivo.com`
`hbk1009@163.com`

## ABSTRACT

Image restoration in real-world conditions is highly challenging due to heterogeneous degradations such as haze, noise, shadows, and blur. Existing diffusion-based methods remain limited: conditional generation struggles to balance fidelity and realism, inversion-based approaches accumulate errors, and posterior sampling requires a known forward operator that is rarely available. We introduce **LearnIR**, a learnable diffusion posterior sampling framework that eliminates this dependency by training a lightweight model to directly predict gradient correction distributions, enabling *Diffusion Posterior Sampling Correction (DPSC)* that maintains consistency with the true image distribution during sampling. In addition, a *Dynamic Resolution Module (DRM)* dynamically adjusts resolution to preserve global structures in early stages and refine fine textures later, while avoiding the need for a pretrained VAE. Experiments on ISTD, O-HAZE, HazyDet, REV-IDE, and our newly constructed FaceShadow dataset show that LearnIR achieves state-of-the-art performance in PSNR, SSIM, and LPIPS.

## 1 INTRODUCTION

Real-world imaging environments often introduce a multitude of degradations—such as haze, noise, shadows, and motion blur—that compromise the visual quality of natural images. Image restoration aims to solve the crucial task of recovering a clean, high-fidelity image from its degraded observation. This is a classic and inherently ill-posed inverse problem that is further complicated by the fact that multiple heterogeneous degradation types often appear simultaneously and are intricately intertwined, thereby posing a significant and persistent challenge for existing restoration methods.

With the development of deep generative models (Ho et al., 2020; Song et al., 2021), a promising direction is to frame image restoration as a conditional image generation problem. By leveraging a degraded image as a condition, these models are trained to synthesize a high-quality image while preserving its underlying structure and semantics. Among these, diffusion models have recently shown unprecedented success, thanks to their capacity to learn intricate distribution transformations. When combined with dedicated mechanisms for low-frequency structure alignment (e.g., resampling (Choi et al., 2021; Cao et al., 2025) or frequency-domain projection (Xiao et al., 2024; Shang et al., 2024)) and high-frequency detail synthesis (e.g., dynamic degradation-aware diffusion (Wang et al., 2025b)), diffusion models can achieve superior performance. Nevertheless, a fundamental trade-off remains: balancing faithful restoration and plausible generation.

A complementary approach to conditional generation is diffusion-inversion-based restoration (Whang et al., 2022; Choi et al., 2021; Wang et al., 2025b; Guo et al., 2023b). While this

---

*Equal contributions. †Corresponding author. Project page: `github.com/gityihang/LearnIR`

paradigm can achieve impressive detail recovery, the process of inverting the degraded image into the latent space inherently accumulates errors, leading to noticeable deviations from the input. Moreover, these methods are often inefficient due to the necessary adding and subsequent removal of noise. In contrast, diffusion-based posterior sampling (Chung et al., 2023; Zhao et al., 2024; Zirvi et al., 2024; Rout et al., 2023) has shown that combining the generative prior with a data consistency constraint can effectively solve ill-posed inverse problems, such as Gaussian deblurring. However, this approach is severely limited in practical and diverse real-world scenarios, since it requires a precisely known forward measurement operator $\mathcal{A}$ (e.g., randomly masking the image proposed in DPS (Chung et al., 2023)), which must also be explicitly reused during inference.

In this work, we introduce LearnIR, a novel learnable diffusion posterior sampling framework for efficient and faithful real-world image restoration. Unlike existing approaches that rely on a known forward measurement operator, our method bypasses this requirement. Leveraging the closure property of the normal distribution, we train a lightweight model to directly predict the distribution of the gradient correction term — a key component in diffusion posterior sampling (Chung et al., 2023) — thereby establishing a robust mechanism to maintain denoising consistency with the true image distribution. Specifically, LearnIR performs Diffusion Posterior Sampling Correction (DPSC) during inference through a compact pre-trained model that adaptively learns and applies corrections to the diffusion trajectory at each step. By removing the dependency on an forward operator, DPSC effectively steers the generative process toward more accurate reconstructions, akin to inversion-based methods, but without requiring an explicit mapping between clean and degraded images.

Furthermore, to obtain richer global context and simplify the pipeline, we introduce a novel Dynamic Resolution Module (DRM) specifically designed for pixel-space image restoration. The DRM dynamically adjusts image resolution during training by applying large-scale downscaling in the early sampling stages to better preserve global structures, followed by large-scale upscaling in the later stages to promote fine texture generation. This design not only achieves a superior balance between structural integrity and detail fidelity but also removes the need for a pretrained VAE, significantly reducing computational costs and enabling a simplified end-to-end image generation pipeline.

We validate the proposed LearnIR framework on the ISTD (Wang et al., 2018), O-HAZE (Ancuti et al., 2018), HazyDet (Ancuti et al., 2018), REVIDE (Zhang et al., 2021) and our newly constructed FaceShadow dataset for the challenging tasks of haze removal and shadow removal. Experimental results clearly demonstrate that LearnIR uniformly outperforms existing methods, including Resfusion (Shi et al., 2024), ShadowRefiner (Dong et al., 2024), and ConvIR (Cui et al., 2024), among others, by a significant margin in PSNR (Hore & Ziou, 2010) and SSIM (Wang et al., 2004) across all tested datasets. These results validate the effectiveness of our proposed LearnIR framework.

## 2 PRELIMINARIES

### 2.1 LEARNING RESIDUAL DENOISING MODELS

Standard inversion-based diffusion methods often suffer from instability in intermediate states, complicating the identification of an optimal sampling starting point. To address this, recent works such as ResFusion (Shi et al., 2024) incorporate a residual term $\boldsymbol{R}$ into the diffusion process, defined as:

$$\boldsymbol{R} = \boldsymbol{y} - \boldsymbol{x}_0,$$

where $\boldsymbol{y}$ represents the degraded image and $\boldsymbol{x}_0$ denotes the clean image. Incorporating this residual into the forward diffusion process modifies the transition equation to:

$$\boldsymbol{x}_t = \sqrt{\bar{\alpha}_t}\,\boldsymbol{x}_0 + \left(1 - \sqrt{\bar{\alpha}_t}\right)\boldsymbol{R} + \sqrt{1 - \bar{\alpha}_t}\,\boldsymbol{\epsilon}, \qquad \boldsymbol{\epsilon} \sim \mathcal{N}(\boldsymbol{0}, \boldsymbol{I}). \tag{1}$$

Substituting $\boldsymbol{R} = \boldsymbol{y} - \boldsymbol{x}_0$ into Eq. equation 1 yields the reformulated forward process:

$$\boldsymbol{x}_t = (2\sqrt{\bar{\alpha}_t} - 1)\boldsymbol{x}_0 + \left(1 - \sqrt{\bar{\alpha}_t}\right)\boldsymbol{y} + \sqrt{1 - \bar{\alpha}_t}\,\boldsymbol{\epsilon}. \tag{2}$$

A critical challenge in the forward process is that $\boldsymbol{x}_0$ is unknown during inference. However, Eq. equation 2 reveals that the dependence on $\boldsymbol{x}_0$ vanishes when the coefficient $(2\sqrt{\bar{\alpha}_t} - 1)$ approaches zero. Based on this observation, a stable sampling timestep $T'$ can be identified via the following minimization (Shi et al., 2024):

$$T' = \arg\min_{i=1}^{T} \left| \sqrt{\bar{\alpha}_i} - \tfrac{1}{2} \right|. \tag{3}$$

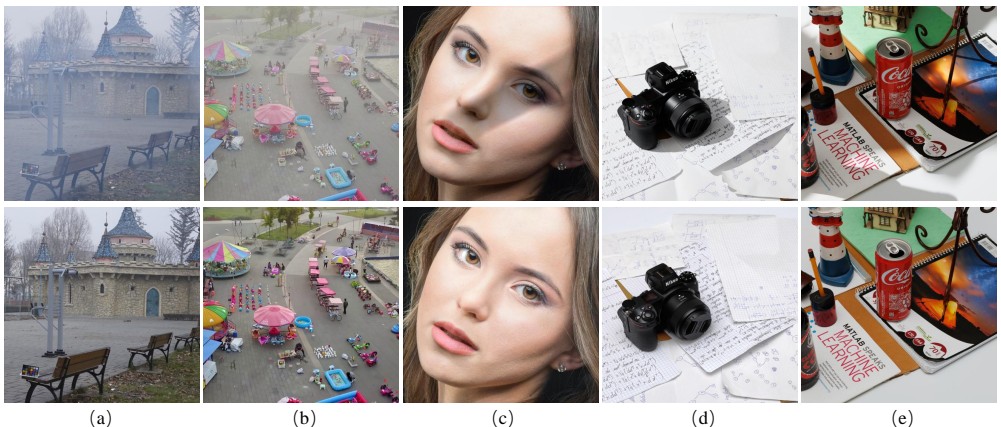

| (a) | (b) | (c) | (d) | (e) |

Figure 1: **Real-World Image Restoration with LearnIR.** This figure provides a visual comparison of LearnIR's real-world image restoration capabilities. The top row shows the degraded inputs, which include various complex degradations like haze and shadows. From left to right, the images show (a) and (b) de-hazing, (c) facial shadow removal, and (d) and (e) object shadow removal. The bottom row presents the corresponding outputs generated by our LearnIR framework. As clearly evidenced by the results, our method effectively removes these complex and intertwined degradations while faithfully preserving fine image details and generating visually high-fidelity results.

At this specific timestep $T'$, the forward state $\boldsymbol{x}_{T'}$ becomes independent of $\boldsymbol{x}_0$ and can be approximated solely using the degraded input $\boldsymbol{y}$ and noise:

$$\boldsymbol{x}_{T'} \approx (1 - \sqrt{\bar{\alpha}_{T'}})\,\boldsymbol{y} + \sqrt{1 - \bar{\alpha}_{T'}}\,\boldsymbol{\epsilon} \approx \sqrt{\bar{\alpha}_{T'}}\,\boldsymbol{y} + \sqrt{1 - \bar{\alpha}_{T'}}\,\boldsymbol{\epsilon}. \tag{4}$$

Consequently, the reverse sampling process can be initiated at $T'$. The model is trained to predict the residual-shifted noise, with the objective function reformulated as:

$$\mathcal{L} = \mathbb{E}_{\boldsymbol{x}_0, \boldsymbol{\epsilon}, t}\left[\|\boldsymbol{\epsilon}^{\mathrm{res}} - \boldsymbol{\epsilon}_\theta^{\mathrm{res}}(\boldsymbol{x}_t, t)\|_2^2\right], \quad \text{where}\;\; \boldsymbol{\epsilon}^{\mathrm{res}} = \boldsymbol{\epsilon} + \frac{(1 - \sqrt{\bar{\alpha}_t})\sqrt{1 - \bar{\alpha}_t}}{\beta_t}\,\boldsymbol{R}. \tag{5}$$

## 2.2 DIFFUSION POSTERIOR SAMPLING FOR NOISY INVERSE PROBLEMS

Inverse problems are fundamental to numerous scientific and engineering applications (Groetsch & Groetsch, 1993), aiming to recover an unknown signal $\boldsymbol{x} \in \mathbb{R}^d$ from a degraded observation $\boldsymbol{y} \in \mathbb{R}^n$ via the model:

$$\boldsymbol{y} = \mathcal{A}(\boldsymbol{x}) + \boldsymbol{n}, \quad \boldsymbol{n} \sim \mathcal{N}(\boldsymbol{0}, \sigma^2 \boldsymbol{I}), \tag{6}$$

where $\mathcal{A}(\cdot)$ is a forward operator (potentially non-linear or ill-posed). Since the mapping $\boldsymbol{x} \to \boldsymbol{y}$ is typically many-to-one, recovering $\boldsymbol{x}$ requires sampling from the intractable posterior distribution $p(\boldsymbol{x} \mid \boldsymbol{y})$.

Diffusion models have recently emerged as robust generative priors for such ill-posed problems. Methods like Diffusion Posterior Sampling (DPS) (Chung et al., 2023) enable conditional sampling by modifying the reverse diffusion score. By applying Bayes' rule, the conditional score function decomposes into:

$$\nabla_{\boldsymbol{x}_t} \log p(\boldsymbol{x}_t \mid \boldsymbol{y}) = \nabla_{\boldsymbol{x}_t} \log p(\boldsymbol{x}_t) + \nabla_{\boldsymbol{x}_t} \log p(\boldsymbol{y} \mid \boldsymbol{x}_t). \tag{7}$$

Here, the prior score $\nabla_{\boldsymbol{x}_t} \log p(\boldsymbol{x}_t)$ is provided by a pretrained diffusion model $s_\theta(\boldsymbol{x}_t, t)$. The second term, the likelihood score $\nabla_{\boldsymbol{x}_t} \log p(\boldsymbol{y} \mid \boldsymbol{x}_t)$, enforces data consistency. However, deriving $p(\boldsymbol{y} \mid \boldsymbol{x}_t)$ analytically is intractable because it requires marginalizing over $\boldsymbol{x}_0$.

To approximate this term, DPS leverages the expected denoised estimate $\hat{\boldsymbol{x}}_0(\boldsymbol{x}_t) = \mathbb{E}[\boldsymbol{x}_0 \mid \boldsymbol{x}_t]$, yielding the approximation:

$$\begin{aligned} p(\boldsymbol{y} \mid \boldsymbol{x}_t) &= \mathbb{E}_{\boldsymbol{x}_0 \sim p(\boldsymbol{x}_0 \mid \boldsymbol{x}_t)}\left[p(\boldsymbol{y} \mid \boldsymbol{x}_0)\right] \\ &\simeq p(\boldsymbol{y} \mid \mathbb{E}[\boldsymbol{x}_0 \mid \boldsymbol{x}_t]) \\ &\simeq p(\boldsymbol{y} \mid \hat{\boldsymbol{x}}_0(\boldsymbol{x}_t)). \end{aligned} \tag{8}$$

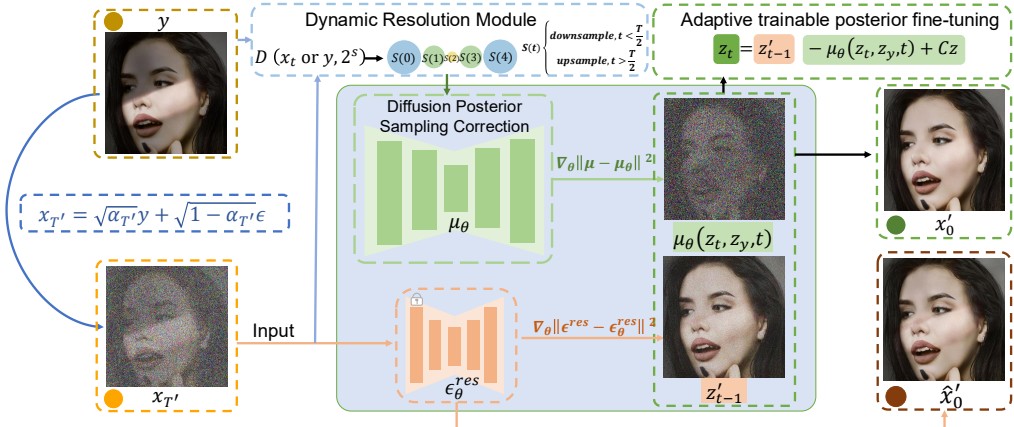

Figure 2: **Overview of the LearnIR Framework.** The smooth equivalence transformation (Shi et al., 2024) identifies the optimal sampling step $T'$ (blue line) to bypass the unstable intermediate states derived from observation $\boldsymbol{y}$. The proposed framework consists of two key components: 1) The **Dynamic Resolution Module (DRM)** projects inputs into a time-dependent latent space via $\mathcal{D}(\cdot, 2^s)$ to balance structural coherence and detail recovery; 2) The **Diffusion Posterior Sampling Correction (DPSC)** acts as a plug-and-play regularization to analytically eliminate the trajectory inconsistency (structural bias) discussed in Appendix A.4.

assuming a Gaussian measurement noise model as defined in Eq. equation 9:

$$p(\boldsymbol{y} \mid \boldsymbol{x}_0) \propto \exp\left(-\frac{1}{2\sigma^2} \|\boldsymbol{y} - \mathcal{A}(\boldsymbol{x}_0)\|_2^2\right). \tag{9}$$

By differentiating the approximated likelihood with respect to $\boldsymbol{x}_t$, we obtain the guidance term:

$$\nabla_{\boldsymbol{x}_t} \log p(\boldsymbol{y} \mid \boldsymbol{x}_t) \simeq -\frac{1}{\sigma^2} \nabla_{\boldsymbol{x}_t} \|\boldsymbol{y} - \mathcal{A}(\hat{\boldsymbol{x}}_0(\boldsymbol{x}_t))\|_2^2.$$

Substituting this back into Eq. equation 7 results in the final conditional score update rule:

$$\nabla_{\boldsymbol{x}_t} \log p(\boldsymbol{x}_t \mid \boldsymbol{y}) \simeq s_\theta(\boldsymbol{x}_t, t) - \rho \nabla_{\boldsymbol{x}_t} \|\boldsymbol{y} - \mathcal{A}(\hat{\boldsymbol{x}}_0)\|_2^2, \quad \text{with} \quad \rho \triangleq \frac{1}{\sigma^2}.$$

## 3 METHODOLOGY

In this section, we present **LearnIR**, a robust framework for residual-based image restoration. As illustrated in Fig. 2, our approach integrates two complementary mechanisms: a **Dynamic Resolution Module (DRM)** that constructs a resolution-aware latent space to suppress noise and redundancy, and a **Diffusion Posterior Sampling Correction (DPSC)** that operates within this space to align the diffusion trajectory with the true posterior. The full algorithm is detailed in Section 3.3.

### 3.1 DRM: DYNAMIC RESOLUTION MODULE

Drawing inspiration from multi-scale generation frameworks such as MDM (Gu et al., 2024) and Pixelflow (Chen et al., 2025), we incorporate a Dynamic Resolution Module (DRM) to enable adaptive processing across spatial scales. By processing high-noise states at lower resolutions, DRM effectively captures global context while mitigating artifacts such as texture inconsistency.

**Resolution-Aware Latent Mapping.** We define a dynamic downsampling operator $\mathcal{D}(\cdot, s)$, where the scale factor $s$ is governed by a time-dependent schedule $s(t)$. Unlike standard diffusion operating in a fixed pixel space, our process evolves in a variable-resolution latent space. Specifically, at timestep $t$, the clean image $\boldsymbol{x}_0$ and the degraded image $\boldsymbol{y}$ are mapped to their latent counterparts:

$$\boldsymbol{z}_0^{(t)} = \mathcal{D}(\boldsymbol{x}_0, s(t)), \quad \boldsymbol{z}_y^{(t)} = \mathcal{D}(\boldsymbol{y}, s(t)). \tag{10}$$

Consequently, $z^{(t)}$ denotes the latent noisy at the specific resolution determined by $s(t)$.

**Latent Residual Diffusion.** To maintain consistency with the residual learning objective (Sec. 2.1), we extend the residual formulation into this dynamic latent space. We define the *latent residual* $\boldsymbol{R_z}$ as:

$$\boldsymbol{R_z} = \boldsymbol{z}_y^{(t)} - \boldsymbol{z}_0^{(t)}. \tag{11}$$

Substituting this into the forward process, the transition kernel in the latent space becomes:

$$q(\boldsymbol{z}_t \mid \boldsymbol{z}_0^{(t)}) = \mathcal{N}\left(\boldsymbol{z}_t;\ \sqrt{\bar{\alpha}_t}\,\boldsymbol{z}_0^{(t)} + (1 - \sqrt{\bar{\alpha}_t})\,\boldsymbol{R_z},\ (1 - \bar{\alpha}_t)\mathbf{I}\right). \tag{12}$$

This ensures that the degradation prior $\boldsymbol{R_z}$ effectively guides the noise injection regardless of the spatial scale.

**Coarse-to-Fine Sampling Strategy.** The schedule $s(t)$ facilitates a coarse-to-fine trajectory, as illustrated in Figure 2. During the early high-noise stages ($t \geq T/2$), the model operates at a downsampled resolution ($s = s_{\text{down}}$) to focus on global structure. In later stages ($t < T/2$), it transitions to the native scale ($s = s_{\text{up}}$) to refine high-frequency details. We implement this resizing via efficient, non-trainable interpolation, allowing LearnIR to leverage pre-trained priors without complex architectural changes.

## 3.2 DPSC: Diffusion Posterior Sampling Correction

**Motivation.** We start from the standard training objective of diffusion models. Typically, the denoising loss is formulated as an $\ell_1$ or $\ell_2$ loss:

$$\mathcal{L}_{\text{denoise}} = \mathbb{E}_{\boldsymbol{z}_0^{(t)}, \boldsymbol{\epsilon}, t} \left\| \boldsymbol{\epsilon} - \epsilon_\theta(\boldsymbol{z}_t, t) \right\|_2^2, \tag{13}$$

where $\epsilon_\theta$ denotes the noise prediction network. While effective, this loss only ensures accurate noise estimation at each timestep, but does not guarantee that the learned reverse posterior $p_\theta(\boldsymbol{z}_{t-1} \mid \boldsymbol{z}_t)$ matches the true posterior $q(\boldsymbol{z}_{t-1} \mid \boldsymbol{z}_t, \boldsymbol{z}_0^{(t)})$ induced by the forward process. This mismatch, which we refer to as *inconsistency*, tends to accumulate across multiple sampling steps and manifests as artifacts such as color shifts.

**Consistency Regularization.** To address this issue, we introduce a consistency regularization term $\mathcal{L}_{\text{consistency}}$, which directly minimizes the discrepancy between the forward-process distribution and the reverse-process distribution predicted by the model. The final training objective is given by

$$\mathcal{L}_{\text{total}} = \mathcal{L}_{\text{denoise}} + \lambda \mathcal{L}_{\text{consistency}}, \tag{14}$$

where $\lambda$ is a weighting hyperparameter.

**Towards Consistency Loss.** To construct $\mathcal{L}_{\text{consistency}}$, we first formally quantify the discrepancy between the forward and reverse processes in the DRM latent space. We achieve this by analyzing the true forward posterior distribution (Definition 1) and the model-predicted reverse distribution (Definition 2). Here, $\boldsymbol{z}_t$ denotes the latent state at timestep $t$, whose spatial resolution is implicitly determined by the schedule $s(t)$ (Sec. 3.1); similarly, $\boldsymbol{z}_0^{(t)}$ and $\boldsymbol{z}_y^{(t)}$ are the resolution-dependent latents of the clean image and the degraded image, respectively.

**Definition 1.** *In the residual formulation of the forward process in Eq. 12, the variable $\boldsymbol{z}_{t-1}^{\text{forward}}$ follows a Gaussian distribution:*

$$\boldsymbol{z}_{t-1}^{\text{forward}} \sim \mathcal{N}\left(\sqrt{\bar{\alpha}_{t-1}}\,\boldsymbol{z}_0^{(t)} + (1 - \sqrt{\bar{\alpha}_{t-1}})\,\boldsymbol{R_z},\ (1 - \bar{\alpha}_{t-1})\,\mathbf{I}\right), \tag{15}$$

*where $\boldsymbol{R}_z = \boldsymbol{z}_y^{(t)} - \boldsymbol{z}_0^{(t)}$ denotes the residual in the DRM latent space.*

**Definition 2.** *The model-predicted reverse distribution $p_\theta(\boldsymbol{z}_{t-1} \mid \boldsymbol{z}_t, \boldsymbol{z}_y^{(t)}, t)$ in the residual-based VP-SDE or DDPM is a Gaussian distribution:*

$$\boldsymbol{z}_{t-1}^{\text{pred}} \sim \mathcal{N}\left(\frac{1}{\sqrt{\alpha_t}}\,\boldsymbol{z}_t - \frac{1 - \alpha_t}{\sqrt{\alpha_t(1 - \bar{\alpha}_t)}}\,\boldsymbol{\epsilon}_\theta^{res}(\boldsymbol{z}_t, \boldsymbol{z}_y^{(t)}, t),\ \frac{(1 - \bar{\alpha}_{t-1})(1 - \alpha_t)}{1 - \bar{\alpha}_t}\,\mathbf{I}\right), \tag{16}$$

*where $\boldsymbol{\epsilon}_\theta^{res}$ denotes the residual noise predictor.*

The stochasticity of the reverse sampling process defined by Eq. 16 introduces an additional sampling noise $\boldsymbol{\xi} \sim \mathcal{N}(\mathbf{0}, \mathbf{I})$. While the injection of this random noise is essential for generating sample diversity, its accumulation across steps may lead to noticeable discrepancies from the desired posterior trajectory, such as color shifts (Appendix A.4). To mitigate this issue, prior works introduce gradient-based corrections, which we briefly review below.

**Definition 3.** (*Denoising Posterior Sampling (Chung et al., 2023)). This method approximates* $p(\boldsymbol{z}_y^{(t)} \mid \boldsymbol{z}_t)$ *by* $p\Big(\boldsymbol{z}_y^{(t)} \mid \hat{\boldsymbol{z}}_0^{(t)} := \mathbb{E}[\boldsymbol{z}_0^{(t)} \mid \boldsymbol{z}_t]\Big)$ *and further estimates*

$$\nabla_{\boldsymbol{z}_t} \log p(\boldsymbol{z}_y^{(t)} \mid \boldsymbol{z}_t) \approx -\frac{1}{\sigma^2} \nabla_{\boldsymbol{z}_t} \left\| \boldsymbol{z}_y^{(t)} - \mathcal{A}\big(\hat{\boldsymbol{z}}_0^{(t)}(\boldsymbol{z}_t)\big) \right\|_2^2, \tag{17}$$

*where* $\mathcal{A}(\cdot)$ *denotes the forward degradation operator in the corresponding space, and the full derivation is provided in Appendix A.10. At each sampling step, the trajectory is corrected as*

$$\boldsymbol{z}_{t-1} \leftarrow \boldsymbol{z}_{t-1}' - \eta \, \nabla_{\boldsymbol{z}_t} \log p(\boldsymbol{z}_y^{(t)} \mid \boldsymbol{z}_t), \tag{18}$$

*where* $\boldsymbol{z}_{t-1}'$ *is the original reverse-diffusion prediction, and* $\eta$ *is a scaling factor that controls the magnitude of the gradient correction.*

**Theorem 1.** *Based on Definitions 1, 2 and 3, the DPS gradient in Eq. 17 can be expressed in the DRM latent space as*

$$\nabla_{\boldsymbol{z}_t} \log p(\boldsymbol{z}_y^{(t)} \mid \boldsymbol{z}_t) \propto \boldsymbol{z}_{t-1}^{\text{pred}} - \boldsymbol{z}_{t-1}^{\text{forward}}. \tag{19}$$

*By comparing Eq. 19 with the update rule in Eq. 18, where* $\boldsymbol{z}_{t-1}'$ *corresponds to* $\boldsymbol{z}_{t-1}^{\text{pred}}$ *and* $\boldsymbol{z}_{t-1}$ *corresponds to* $\boldsymbol{z}_{t-1}^{\text{forward}}$*, we see that the DPS gradient precisely captures the discrepancy between the forward and reverse processes in the DRM latent space. Leveraging the closure property of Gaussian distributions, this discrepancy can be modeled as*

$$\boldsymbol{z}_{t-1}^{pred} - \boldsymbol{z}_{t-1}^{forward} \sim \mathcal{N}\Big(\boldsymbol{\mu}(\boldsymbol{z}_t, \boldsymbol{z}_y^{(t)}, t), \; \sigma^2(\boldsymbol{z}_t, \boldsymbol{z}_y^{(t)}, t)\,\mathbf{I}\Big), \tag{20}$$

*where the closed-form expressions of* $\boldsymbol{\mu}$ *and* $\sigma^2$ *are given in Appendix A.11.*

**Final Consistency Loss.** Based on the above derivation, we define the consistency loss as the $\ell_2$ distance between the analytic mean $\boldsymbol{\mu}$ and the predicted mean $\hat{\mu}_\theta$ of the offset:

$$\mathcal{L}_{\text{consistency}} = \mathbb{E}_{\boldsymbol{z}_t, \boldsymbol{z}_y^{(t)}, t} \left\| \boldsymbol{\mu}(\boldsymbol{z}_t, \boldsymbol{z}_y^{(t)}, t) - \hat{\mu}_\theta(\boldsymbol{z}_t, \boldsymbol{z}_y^{(t)}, t) \right\|_2^2, \tag{21}$$

where $\boldsymbol{\mu}$ is analytically computed via Eq. 51, and $\hat{\mu}_\theta$ is the corresponding prediction output by the learnable correction network.

### 3.3 TRAINING AND INFERENCE

**Algorithm Overview.** Algorithms 1 and 2 summarize the training and sampling procedures of LearnIR in the DRM latent space. During training (Algorithm 1), we first determine a truncation step $T'$ via the smooth equivalence transformation, which locates a stable midpoint of the diffusion trajectory. At each iteration, we sample $t \sim \text{Uniform}(\{1, \ldots, T'\})$, and project the clean image $\boldsymbol{x}_0$ and degraded observation $\boldsymbol{y}$ into the resolution-aware latent space via DRM to obtain $\boldsymbol{z}_0^{(t)}$ and $\boldsymbol{z}_y^{(t)}$. Their difference $R_z = \boldsymbol{z}_y^{(t)} - \boldsymbol{z}_0^{(t)}$ encodes the degradation prior at the current scale. Based on $(\boldsymbol{z}_0^{(t)}, R_z)$, we construct the forward latent $\boldsymbol{z}_{t-1}^{\text{forward}}$ with the residual diffusion process and derive a residual noise target $\epsilon^{\text{res}}$ that absorbs the effect of $R_z$. The network predicts both a residual noise $\epsilon_\theta^{\text{res}}$ and a correction mean $\mu_\theta$ that approximates the analytic mean $\mu$ of the discrepancy between the forward and reverse posteriors (cf. Eq. 20). The overall objective is therefore the sum of a denoising loss on $\epsilon_\theta^{\text{res}}$ and a consistency loss that regresses $\mu_\theta$ toward $\mu$, explicitly aligning the learned reverse dynamics with the true forward posterior in the DRM latent space.

At inference time (Algorithm 2), we again compute $T'$ and initialize $\boldsymbol{z}_{T'}$ around the latent observation $\boldsymbol{z}_y^{(T')}$. For each timestep $t = T', T'-1, \ldots, 2$, DRM provides the latent observation $\boldsymbol{z}_y^{(t)}$, upon which we first perform a standard reverse-diffusion update to obtain a provisional state $\boldsymbol{z}_{t-1}'$. Then DPSC is applied: the correction network $\mu_\theta$ together with a Gaussian perturbation $C\boldsymbol{z}$ yields an

**Algorithm 1** Training

**Require:** Total diffusion steps $T$, degraded image $y$, ground truth $x_0$
1: **repeat**
2:    $T' = \arg\min_{i=1}^{T} \left| \sqrt{\alpha_i} - \frac{1}{2} \right|$   // Eq. 22
3:    Sample $t \sim \text{Uniform}(1, \ldots, T')$
4:    $\mathbf{z}_0^{(t)}, \mathbf{z}_y^{(t)} = \mathcal{D}(\mathbf{x}_0, s(t)), \mathcal{D}(\mathbf{y}, s(t))$  // Sec. 3.1
5:    $R_z = \mathbf{z}_y^{(t)} - \mathbf{z}_0^{(t)}$   // Eq. 11
6:    $\mathbf{z}_{t-1}^{\text{forward}} = \sqrt{\bar{\alpha}_{t-1}} \mathbf{z}_0^{(t)} + (1 - \sqrt{\bar{\alpha}_{t-1}}) R_z + \sqrt{1 - \bar{\alpha}_{t-1}} \boldsymbol{\epsilon}$
7:    $\epsilon^{\text{res}} = \epsilon + \frac{(1-\sqrt{\alpha_t})\sqrt{1-\bar{\alpha}_t}}{\beta_t} R_z$
8:    $z_{t-1}^{\text{pred}} = \frac{1}{\sqrt{\alpha_t}} z_t - \frac{\beta_t}{\sqrt{\alpha_t(1-\bar{\alpha}_t)}} \epsilon_\theta^{\text{res}}(\mathbf{z}_t, \mathbf{z}_y^{(t)}, t) + \sqrt{\tilde{\beta}_t} \mathbf{z}$
9:    $\mathbf{z}_{t-1}^{\text{pred}} - \mathbf{z}_{t-1}^{\text{forward}} \sim \mathcal{N}(\mu(\mathbf{z}_t, \mathbf{z}_y^{(t)}, t), C^2 I)$  // Eq.20
10:   Take gradient step on
11:   $\nabla_\theta \left\| \epsilon^{\text{res}} - \epsilon_\theta^{\text{res}}(\mathbf{z}_t, \mathbf{z}_y^{(t)}, t) \right\|^2 + \nabla_\theta \left\| \mu - \mu_\theta(\mathbf{z}_t, \mathbf{z}_y^{(t)}, t) \right\|^2$  // Sec.3.2
12: **until** convergence

**Algorithm 2** Sampling

**Require:** Total steps $T$, degraded image $y$, priors $\{\zeta_i\}_{i=1}^N$, pretrained $\epsilon_{\text{res}}^\theta, \mu_\theta$
1: $\tilde{\beta}_t = \frac{1-\bar{\alpha}_{t-1}}{1-\bar{\alpha}_t} \beta_t$
2: $T' = \arg\min_{i=1}^{T} \left| \sqrt{\alpha_i} - \frac{1}{2} \right|$
3: Sample $\epsilon \sim \mathcal{N}(0, I)$
4: $z_{T'} = \sqrt{\bar{\alpha}_{T'}} \mathbf{z}_y^{(t)} + \sqrt{1 - \bar{\alpha}_{T'}} \epsilon$
5: **for** $t = T', T'-1, \ldots, 2$ **do**
6:   Sample $\mathbf{z} \sim \mathcal{N}(0, I)$
7:   $z'_{t-1} = \frac{1}{\sqrt{\alpha_t}} \left( \mathbf{z}_t - \frac{\beta_t}{\sqrt{1-\bar{\alpha}_t}} \epsilon_\theta^{\text{res}}(\mathbf{z}_t, \mathbf{z}_y^{(t)}, t) \right) + \sqrt{\beta_t} \mathbf{z}$
8:   $\nabla_{\mathbf{z}_t} \log p(\mathbf{z}_y^{(t)} \mid \mathbf{z}_t)^{\text{pred}} = \mu_\theta(\mathbf{z}_t, \mathbf{z}_y^{(t)}, t) + C\mathbf{z}$
9:   $\mathbf{z}_{t-1} = \mathbf{z}'_{t-1} - \nabla_{\mathbf{z}_t} \log p(\mathbf{z}_y^{(t)} \mid \mathbf{z}_t)^{\text{pred}}$
10: **end for**
11:   **return** $x_0 = \frac{1}{\sqrt{\bar{\alpha}_1}} \left( \mathbf{z}_1 - \frac{\beta_1}{\sqrt{1-\bar{\alpha}_1}} \epsilon_\theta^{\text{res}}(\mathbf{z}_1, \mathbf{z}_y^{(t)}) \right) - \mu_\theta(\mathbf{z}_1, \mathbf{z}_y^{(t)})$

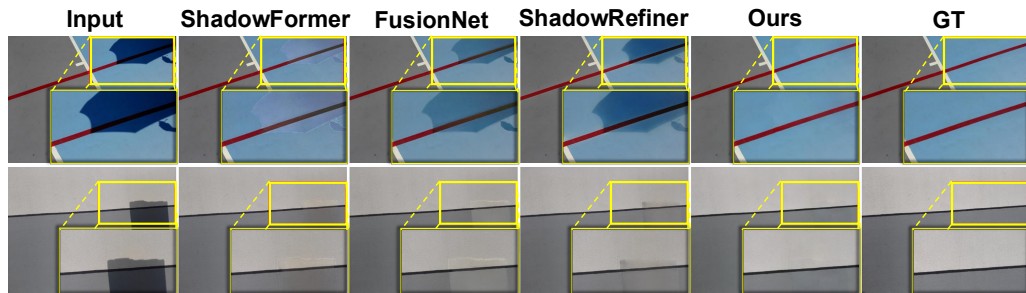

Figure 3: **Visual comparisons on the ISTD datasets**. Qualitative comparison of shadow removal results on the ISTD dataset using different restoration methods.

approximation to the DPS gradient in the latent space, which is subtracted from $\mathbf{z}'_{t-1}$ to produce the corrected state $\mathbf{z}_{t-1}$ that better follows the desired posterior trajectory while respecting the degradation constraint. At $t = 1$, the restored image $\mathbf{x}_0$ is reconstructed from $\mathbf{z}_1$ through the residual-based reverse transform, with an additional correction term from $\mu_\theta$. In this way, DRM controls the spatial scale for robust structure modeling, while DPSC suppresses trajectory inconsistency and structural bias along the entire sampling path.

## 4 EXPERIMENTS

### 4.1 EXPERIMENT SETUP

We conduct experiments on the same benchmarks, including the ISTD dataset (Wang et al., 2018), several dehazing datasets (O-Haze Ancuti et al. (2018), HazyDet Feng et al. (2024), and RE-VIDE Zhang et al. (2021)), and our newly constructed FaceShadow dataset. The detailed construction procedure and implementation details of the FaceShadow dataset are further presented in Appendix A.5.3. Detailed descriptions of the datasets, evaluation protocols, and experimental setups—including training hyperparameters and inference strategies—are provided in Appendix A.5. Since different methods involve distinct model architectures and parameter settings, we adopt their publicly recommended configurations for both training and inference, and perform all evaluations on a single A100 GPU for consistency.

We build upon the RDDM (Liu et al., 2024) architecture to train the residual diffusion model, and introduce an additional branch for posterior refinement. Appendix A.6.1 provides ablation analysis on how different architectural choices affect residual modeling and posterior adjustment performance.

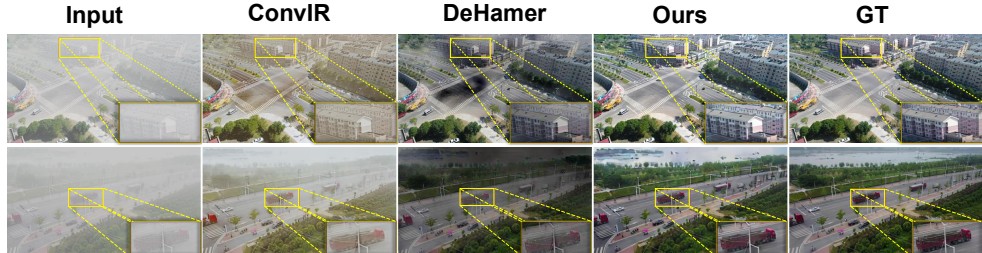

Figure 4: **Visual comparisons on the HazyDet datasets**. Visual comparisons of dehazing results produced by different methods on the HazyDet dataset.

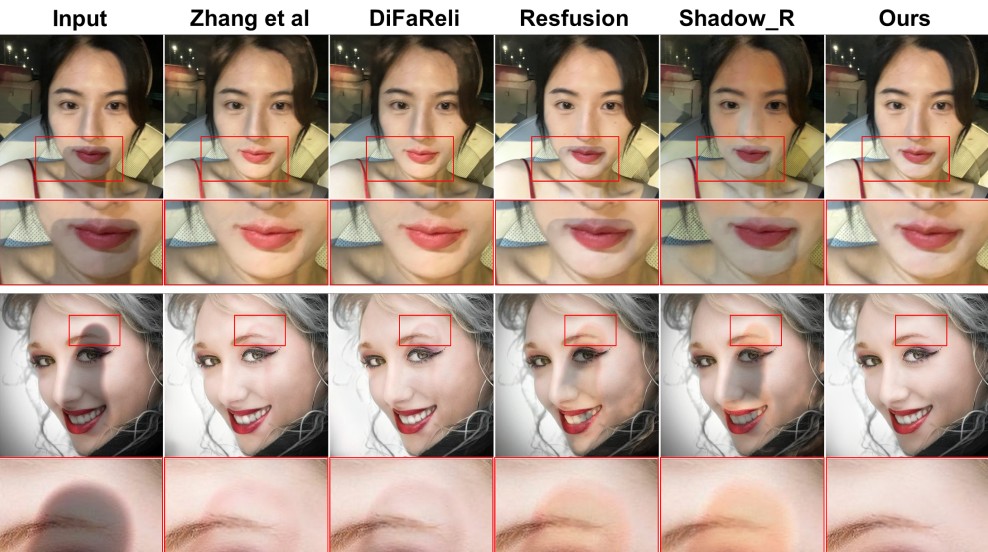

Figure 5: Visual comparisons on 1,000 real-world face shadow images from the FaceShadow test set.

Table 1: Quantitative comparison of shadow removal methods at a resolution of $256 \times 256$ on the ISTD dataset. Methods are categorized based on the use of shadow masks. "Mask-free" indicates that the model does not use explicit shadow masks during training or inference (Dong et al., 2024). **Violet** highlights the best-performing method among mask-free approaches, while **blue** indicates the best among methods using masks. The best and second-best results are marked in **bold** and underline, respectively.

| Methods | Mask-free | PSNR ↑ | SSIM ↑ | MAE ↓ | LPIPS ↓ |
|---|---|---|---|---|---|
| Input Image | - | 20.33 | 0.877 | 11.35 | 0.198 |
| DHAN (Cun et al., 2020) | No | 26.86 | 0.919 | 6.29 | 0.105 |
| FusionNet (Fu et al., 2021) | No | 25.88 | 0.715 | 5.88 | 0.112 |
| ShadowFormer (Guo et al., 2023a) | No | **30.47** | 0.928 | 5.34 | 0.075 |
| Resfusion (Shi et al., 2024) | No | 30.09 | **0.932** | **4.79** | **0.068** |
| DCShadowNet (Jin et al., 2021) | Yes | 24.02 | 0.677 | 5.88 | 0.125 |
| ShadowRefiner (Dong et al., 2024) | Yes | 28.75 | 0.916 | 5.48 | 0.080 |
| Refusion (Luo et al., 2023) | Yes | 25.13 | 0.871 | 5.63 | 0.109 |
| **LearnIR (Ours)** | Yes | **29.57** | **0.927** | **5.12** | **0.072** |

Table 2: Quantitative comparison of various dehazing methods at a fixed resolution of $512 \times 512$ pixels. For evaluation, three widely used benchmark datasets are employed: O-Haze, HazyDet, and REVIDE. In the results table, the best and second-best scores are highlighted in **bold** and underline, respectively.

| Methods | O-HAZE (Ancuti et al., 2018) | | | HazyDet (Feng et al., 2024) | | | REVIDE (Zhang et al., 2021) | | |
|---|---|---|---|---|---|---|---|---|---|
| | PSNR ↑ | SSIM ↑ | LPIPS ↓ | PSNR ↑ | SSIM ↑ | LPIPS ↓ | PSNR ↑ | SSIM ↑ | LPIPS ↓ |
| Input Image | 13.38 | 0.520 | 0.355 | 15.98 | 0.553 | 0.310 | 16.52 | 0.601 | 0.280 |
| FFA-Net (Qin et al., 2020) | 22.12 | 0.770 | 0.150 | 23.11 | 0.726 | 0.145 | 22.80 | 0.735 | 0.160 |
| GridDehazeNet (Liu et al., 2019) | 18.92 | 0.672 | 0.205 | 20.90 | 0.698 | 0.180 | 21.05 | 0.690 | 0.195 |
| MSBDN (Dong et al., 2020) | 24.36 | 0.749 | 0.130 | 24.77 | 0.712 | 0.135 | 24.50 | 0.720 | 0.140 |
| DeHamer (Guo et al., 2022) | 25.11 | 0.777 | 0.115 | 25.39 | 0.767 | 0.110 | 25.15 | 0.765 | 0.115 |
| ConvIR (Cui et al., 2024) | 25.36 | 0.780 | 0.108 | 25.67 | 0.781 | 0.102 | 25.05 | 0.755 | 0.105 |
| Resfusion (Shi et al., 2024) | 24.25 | 0.710 | 0.145 | 24.13 | 0.701 | 0.155 | 24.00 | 0.705 | 0.165 |
| MB-TaylorFormer V2 (Jin et al., 2025) | 25.43 | 0.792 | 0.105 | 24.97 | 0.755 | 0.120 | 25.25 | 0.775 | 0.118 |
| **LearnIR (Ours)** | **27.70** | **0.832** | **0.055** | **27.32** | **0.905** | **0.065** | **25.43** | **0.795** | **0.085** |

## 4.2 QUANTITATIVE RESULTS

**ISTD Dataset.** We compare our approach against DHAN (Cun et al., 2020), FusionNet (Fu et al., 2021), ShadowFormer (Guo et al., 2023a), Resfusion (Shi et al., 2024), DCShadowNet (Jin et al., 2021), ShadowRefiner (Dong et al., 2024), and Refusion (Luo et al., 2023). As shown in Tab. 1, our method outperforms mask-free approaches, with a PSNR gain of 0.82 dB, SSIM improvement of 0.011, and MAE reduction of 0.36, while remaining competitive with mask-based models.

**FaceShadow Dataset.** We benchmark against Zhang et al. (2020), DiFaReli (Ponglertnapakorn et al., 2023), Resfusion (Shi et al., 2024), and ShadowRefiner (Dong et al., 2024). Due to space limitations, some quantitative results are moved to Appendix A.5.4. Tab. 5 shows that our method achieves the best PSNR, SSIM, and LPIPS, with gains of 2.44 dB, 0.073, and 0.013 respectively, highlighting its effectiveness under complex facial shadow conditions.

**Dehazing Datasets.** We compare against several representative dehazing approaches, including FFA-Net (Qin et al., 2020), GridDehazeNet (Liu et al., 2019), MSBDN (Dong et al., 2020), De-Hamer (Guo et al., 2022), ConvIR (Cui et al., 2024), and MB-TaylorFormer V2 (Jin et al., 2025). As shown in Tab. 2, our method consistently outperforms these state-of-the-art models across all three widely used benchmarks—O-HAZE, HazyDet, and REVIDE.

Specifically, our approach achieves significant performance gains of +2.27 dB PSNR and +0.04 SSIM on O-HAZE, and +1.65 dB PSNR and +0.124 SSIM on HazyDet, demonstrating its robustness in real-world hazy scenes. Furthermore, on the REVIDE dataset, our method also obtains superior restoration quality with clear improvements in PSNR, SSIM, and LPIPS, confirming its effectiveness on challenging real-world hazy images.

## 4.3 QUALITATIVE COMPARISONS

Fig. 3, 4, and 5 show qualitative comparisons on the ISTD dataset, the Dehazing datasets, and the FaceShadow dataset. Our results are consistently closer to the ground truth in both color fidelity and structural detail, removing shadows and haze without introducing artifacts or color shifts. Compared with other methods, our pixel-space diffusion formulation better preserves high-frequency details and suppresses boundary artifacts, while maintaining realistic textures and color saturation across diverse scenes. For challenging facial images under complex lighting, our approach further demonstrates stronger structure-preserving capabilities. Additional real-world results are provided in Appendix A.5.7.

## 4.4 ABLATION STUDY

We conduct ablation experiments on the FaceShadow test dataset. Additional results are included in the appendix A.6 for completeness, further validating our method.

Table 3: Ablation study results under different model configurations.

| Configurations | PSNR ↑ | SSIM ↑ | LPIPS ↓ |
|---|---|---|---|
| w/o DPSC | 24.12 | 0.899 | 0.072 |
| w/o DRM | 27.25 | 0.925 | 0.063 |
| w/o DRM & DPSC | 22.86 | 0.865 | 0.103 |
| **Full Model** | **28.52** | **0.965** | **0.058** |

Figure 6: **Ablation Studies**. Visual comparisons on the FaceShadow dataset under different setups.

### 4.4.1 MODULE-WISE ABLATION STUDY

**Dynamic Resolution Module (DRM).**    To evaluate the contribution of the proposed dynamic resolution module, we first remove it and examine its impact on overall performance. As reported in Tab. 3, removing this module results in performance degradation, including a notable drop of 5.66 dB in PSNR, 0.10 in SSIM, and a 0.045 increase in LPIPS. Visual results in Fig. 6 confirm that omitting this component yields inferior reconstructions compared to the full model. Nevertheless, this module allows multi-scale image resizing during training, significantly accelerating convergence. These findings emphasize the importance of MSD in achieving high-quality image restoration.

**Diffusion Posterior Sampling Correction (DPSC).**    From the results in Tab. 3, we observe that removing the DPS module significantly hurts performance—causing a drop of 5.66 dB in PSNR, 0.066 in SSIM, and an increase of 0.045 in LPIPS. Visual comparisons in Fig. 6 further highlight the degradation. Specifically, in the second and third columns of the first row, models without DPS introduce prominent artifacts, suggesting that dynamic offset alone is insufficient. Moreover, using only the diffusion model without DPS leads to residual artifacts that persist in the final result. Our full model yields the cleanest and most faithful reconstructions. These results underscore the critical role of the proposed DPS module in enhancing both visual and perceptual quality.

## 5 CONCLUSION

In this paper, we propose a structure-preserving consistency offset module designed to enhance the quality of image restoration. Specifically, we first provide a theoretical reformulation of the posterior distribution involved in the denoising process during training. This addresses the issue of distributional shift at inference when no explicit reverse process is available. By fitting the inferred posterior distribution during sampling, our approach effectively approximates the real image distribution, thereby improving structural consistency and visual fidelity of the restored outputs. Additionally, we introduce a dynamic resolution module that adaptively adjusts image resolution during training, enabling the model to capture distributional characteristics in the pixel space from multiple scales. This mechanism not only accelerates training but also enhances structural representation and fine-grained detail preservation in the outputs. Extensive experiments across various image restoration tasks demonstrate that our proposed consistency-driven offset mechanism significantly outperforms existing methods in terms of both structural consistency and perceptual quality.

## 6 REPRODUCIBILITY STATEMENT

We are committed to the reproducibility of our theoretical and empirical results. All theoretical claims made in this paper are accompanied by detailed derivations and proofs, which can be found in Appendix A.11. We clearly state all assumptions required for our theorems in Section 2 and 3. Our proposed algorithm is formally described with comprehensive pseudocode in Algorithm 1 and 2 of the main paper. Furthermore, a complete description of the experimental setup, including all model hyperparameters, data processing procedures, and evaluation protocols, is available in Appendix A.5. We believe this provides sufficient information for an independent re-implementation of our work.

## ACKNOWLEDGEMENTS

This work was supported in part by the Guangxi Natural Science Foundation (2024GXNS-FAA010493), the Guangxi Key Research and Development Program (JB2504240003), the Shanghai Municipal Commission of Economy and Informatization (2024-GZL-RGZN-01008), the National Natural Science Foundation of China (52262047, 61862015), and the Science and Technology Project of Guangxi (AD23023002, AD21220114).

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

## A  APPENDIX

### A.1  RELATED WORK IN IMAGE RESTORATION

Previous image restoration methods can be broadly categorized into two paradigms: inversion-based methods and conditional diffusion-based methods.

**Inversion-based Methods.**   Several works (Delbracio & Milanfar, 2023; Li et al., 2025; Wu et al., 2024) enhance diffusion-based image restoration by embedding inversion mechanisms into the generative process, effectively bridging the gap between degraded observations and diffusion priors. To improve alignment between intermediate latents and measurements, recent methods (Shi et al., 2024; Liu et al., 2024; Shang et al., 2024) incorporate residual priors, which mitigate error accumulation and boost reconstruction quality. Moreover, efficient inversion-based solvers (Zirvi et al., 2024; Wang et al., 2025a) have been proposed to enable real-time adaptation in blind restoration. For instance, Chihaoui et al. (2024) optimizes the initial noise input while preserving the sampling trajectory, and accelerates inversion via large time-step skipping. Similarly, Wu et al. (2024) enforce measurement consistency at each step and leverage signal-guided initialization for improved quality with negligible computational overhead.

**Conditional diffusion-based Methods.** Conditional diffusion models have significantly advanced image restoration by incorporating degradation-aware priors, offering more controllable and accurate generation than their unconditional counterparts. Early works such as (Meng et al., 2022) and (Saharia et al., 2022) adopt supervised conditional frameworks, where degraded images are introduced either as direct inputs or as intermediate guidance during sampling. Building on this idea, Plug-and-Play approaches, including (Kawar et al., 2022), (Zhu et al., 2023), and (Chung et al., 2023), decouple training and inference by guiding pretrained unconditional priors with degraded observations, without altering the backbone network. To further improve efficiency and generalization, unified conditional modeling frameworks have emerged. For example, (Zhang et al., 2023), (Mei et al., 2024), and (Rajagopalan et al., 2025) incorporate encoded degradation features into the diffusion process, with GenDeg further synthesizing diverse corruptions to enhance robustness against unknown degradations. In addition, several task-specific architectures have been developed to target distinct degradation types: Yue et al. (2025) jointly models atmospheric scattering and degradation masks for accurate dehazing, Luo et al. (2023) employs a dual-domain Transformer to address both low-light and blur conditions, and Ye et al. (2024) enhances high-frequency details via learned texture priors.

## A.2 Use of Large Language Models (LLMs)

In the preparation of this manuscript, Large Language Models (LLMs) were utilized as an auxiliary tool for language refinement and stylistic improvement. Specifically, these models assisted in enhancing clarity, grammar, coherence, and conciseness across various sections of the paper. This application of LLMs was strictly limited to editorial support—such as rephrasing sentences, identifying grammatical errors, and suggesting more academic phrasing—and did not involve content generation, conceptual development, or the formulation of scientific arguments. The authors maintained full intellectual control over the scientific content and conclusions presented herein.

## A.3 Proof Following Resfusion

For Eq. 15, Resfusion (Shi et al., 2024) introduces the residual term $R$ and identifies an optimal sampling step $T'$ that minimizes information loss. This step corresponds to the point where $\sqrt{\alpha_t}$ is closest to $\frac{1}{2}$:

$$T' = \underset{i=1}{\overset{T}{\arg\min}} \left| \sqrt{\alpha_i} - \frac{1}{2} \right|. \tag{22}$$

At this step, $x_{T'}$ can be approximately represented in two equivalent forms:

$$\begin{aligned} z_{T'} &\approx (1 - \sqrt{\alpha_{T'}}) D^s(y) + \sqrt{1 - \alpha_{T'}}\, \epsilon \\ &\approx \sqrt{\alpha_{T'}}\, D^s(y) + \sqrt{1 - \alpha_{T'}}\, \epsilon. \end{aligned} \tag{23}$$

Here, $\epsilon \sim \mathcal{N}(0, I)$ denotes standard Gaussian noise, and $y$ is the input to the model. These approximations suggest that at the specific step $T'$, the intermediate variable $x_{T'}$ preserves both the structural content of $y$ and stochastic perturbations, making it a high-fidelity intermediate representation.

## A.4 Structural Bias in Residual-Guided Diffusion Models

In residual-guided diffusion models, the forward process is reformulated as:

$$x_t = \sqrt{\alpha_t}\, x_{t-1} + (1 - \sqrt{\alpha_t})\, R + \sqrt{1 - \alpha_t}\, \epsilon, \tag{24}$$

where $x_0$ is the ground truth, $R = \hat{x}_0 - x_0$ is the degradation residual, and $\epsilon \sim \mathcal{N}(0, I)$ is Gaussian noise.

The model jointly predicts residual and noise components,

$$\epsilon_\theta^{\text{res}} = \hat{\epsilon}_\theta(x_t, t) + \hat{R}_\theta(x_t, t),$$

leading to the total training loss:

$$\mathcal{L}_{\text{total}} = \mathbb{E}_{x_0, \epsilon, t} \left[ \|\hat{\epsilon}_\theta(x_t, t) - \epsilon\|^2 + \lambda \|\hat{R}_\theta(x_t, t) - R\|^2 \right]. \tag{25}$$

Since $R$ and $\epsilon$ are generally correlated ($\text{Cov}(R, \epsilon) \neq 0$), their estimation interferes with each other.

**Proposition 1** (Mutual Estimation Interference). *Let $\boldsymbol{x}_t = A\boldsymbol{R} + B\boldsymbol{\epsilon}$ with $A = 1 - \sqrt{\alpha_t}$, $B = \sqrt{1 - \alpha_t}$, and assume $\boldsymbol{R}, \boldsymbol{\epsilon}$ are correlated. Then no estimator $\theta$ can simultaneously minimize*

$$\mathbb{E}[\|\hat{\boldsymbol{R}}_\theta(\boldsymbol{x}_t) - \boldsymbol{R}\|^2] + \mathbb{E}[\|\hat{\boldsymbol{\epsilon}}_\theta(\boldsymbol{x}_t) - \boldsymbol{\epsilon}\|^2]$$

*without interference from cross terms induced by* $\text{Cov}(\boldsymbol{R}, \boldsymbol{\epsilon}) \neq 0$.

**Implication.** Noise and residual jointly determine the perturbation direction of $\boldsymbol{x}_t$, so their estimations inherently compete. Reducing error in one component necessarily increases error in the other, leading to a fundamental trade-off.

## A.5 EXPERIMENTAL DETAILS AND DATASETS

### A.5.1 EXPERIMENTAL DETAILS

We use the peak signal-to-noise ratio (PSNR), structural similarity (SSIM, (Wang et al., 2004)), learned perceptual image patch similarity (LPIPS, (Zhang et al., 2018)), and mean absolute error (MAE) as quantitative metrics. The batch size, image resolution, and training epochs for each task are summarized in Tab. 4.

Table 4: Experimental settings for our LearnIR during the training stage

| Tasks | Shadow Removal | | Haze Removal | | | Super-Resolution |
|---|---|---|---|---|---|---|
| | ISTD | FaceShadow | O-HAZE | HazyDet | REVIDE | DIV2K |
| Batch size | 32 | 16 | 16 | 16 | 16 | 16 |
| Image/patch size | 256 | 512 | 512 | 512 | 512 | 1024 |
| Sampling steps | 5 | 5 | 5 | 5 | 5 | 5 |
| Learning rate | 1e-4 | 1e-4 | 1e-4 | 1e-4 | 1e-4 | 1e-4 |
| Training epochs | 5k | 5k | 5k | 5k | 5k | 5k |

### A.5.2 TRAINING DETAILS

We adopt a two-stage training strategy for LearnIR. In **Stage 1**, we train LearnIR at a fixed input resolution, with the Dynamic Resolution Module (DRM) disabled, so that the model first focuses on learning the gradient correction term for Diffusion Posterior Sampling Correction (DPSC). In **Stage 2**, we fine-tune the same model with DRM enabled under dynamic resolutions using a smaller learning rate, allowing LearnIR to quickly adapt to various image resolutions while preserving the DPSC behavior learned in Stage 1.

### A.5.3 DATASETS

**ISTD Dataset.** The **ISTD** (Wang et al., 2018) is a widely used benchmark for shadow removal. It consists of triplets of shadow images, corresponding shadow masks, and ground-truth shadow-free images. The dataset includes 1,330 triplets for training and 540 triplets for quantitative evaluation.

In real-world applications, shadow images typically lack corresponding annotated masks, and manually creating high-quality masks is expensive and hard to scale. Our proposed consistency offset mechanism can effectively leverage information from pretrained data to enable high-quality shadow removal without requiring shadow masks during inference.

**Dehazing datasets.** **O-Haze** (Ancuti et al., 2018) is the first outdoor haze dataset consisting of 45 pairs of hazy and corresponding haze-free images, collected across diverse outdoor scenes. The hazy images are captured under real atmospheric conditions using professional haze machines, ensuring photorealistic haze simulation. Importantly, both hazy and clear images are taken under identical illumination settings in controlled environments. **HazyDet** (Feng et al., 2024) is a large-scale dataset containing over 383,000 real-world instances, constructed from synthetic hazy images generated using the Atmospheric Scattering Model (ASM). The hazy variants are derived from clear-weather UAV images, making the dataset suitable for evaluating real-world dehazing performance. **REVIDE** (Zhang et al., 2021) is a real-world video dehazing dataset constructed from diverse outdoor scenes with naturally captured haze conditions. It provides paired hazy and haze-free video

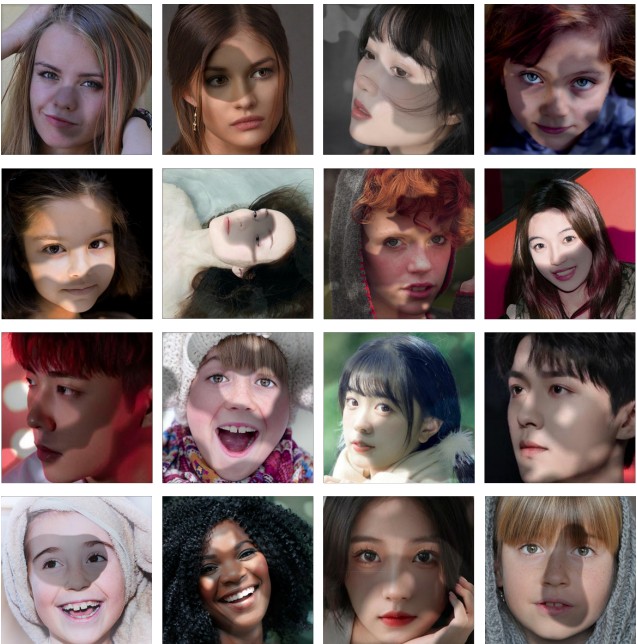

Figure 7: **Synthetic Face Shadow Dataset** Samples from our synthesized FaceShadow dataset, which includes diverse identities and scene conditions with realistic facial shadows.

sequences, enabling frame-level supervision for evaluating dehazing performance under dynamic illumination, motion, and atmospheric variations. The dataset contains high-quality annotations and consistent temporal correspondence, making it a valuable benchmark for assessing both image- and video-level dehazing robustness.

**FaceShadow Dataset.** In real-world scenarios, facial images are often affected by various lighting conditions, particularly shadows. These interferences not only degrade the accuracy of computer vision tasks such as face recognition and expression analysis but also impose greater challenges on the generalization ability of related models. However, existing public face datasets contain extremely limited paired data with explicit shadow and shadow-free correspondences, making it difficult to obtain accurate one-to-one real-world samples. Therefore, to better study and advance the task of face shadow removal, it is essential to construct a large-scale, high-quality, and paired synthetic face shadow dataset. To this end, we build upon the face shadow synthesis method proposed by (Zhang et al., 2020) to construct a novel dataset FaceShadow which contains 30,000 image pairs. Each pair consists of a shadowed facial image and its corresponding shadow-free version. The dataset features rich diversity across different ages, genders, and scene conditions, as illustrated in Fig. 7. The 30,000 synthetic pairs are split into **29,000 pairs for training** and 1,000 pairs for quantitative evaluation on the validation set. Furthermore, to thoroughly evaluate the generalization of our proposed method in real-world applications, we introduce an additional **1,000 pairs of real face shadow images collected from the Internet** to serve as the test set for fine-tuning and real-world application testing.

**DIV2K Dataset Evaluation.** The **DIV2K** (Agustsson & Timofte, 2017) (Diverse 2K resolution images for image super-resolution) is a critical benchmark in the SR field, consisting of 1000 high-resolution (2K) diverse images, commonly used for training and comprehensive testing of modern SR models. Due to its size and image complexity, performance on DIV2K is often considered the definitive measure of a model's robustness and generalization ability.

### A.5.4 ADDITIONAL EXPERIMENTAL RESULTS

For completeness, we provide additional quantitative comparisons in terms of PSNR, SSIM, and MAE on the FaceShadow datasets. These detailed results are presented in Tabs. 5 due to space limitations in the main text.

Table 5: Quantitative comparison of face shadow removal methods under $512 \times 512$ resolution. Two datasets are used for evaluation: the synthetic FaceShadow dataset and a real-world test set. The best and second-best results are highlighted in **bold** and underline, respectively.

| Methods | Synthetic Dataset | | | Real Dataset | |
|---|---|---|---|---|---|
| | PSNR ↑ | SSIM ↑ | LPIPS ↓ | PSNR ↑ | SSIM ↑ |
| Input Image | 16.28 | 0.823 | 0.159 | 17.12 | 0.825 |
| Zhang et al. (Zhang et al., 2020) | 25.18 | 0.876 | 0.062 | 24.32 | 0.865 |
| DiFaReli (Ponglertnapakorn et al., 2023) | 26.43 | 0.897 | 0.073 | 25.48 | 0.872 |
| Resfusion (Shi et al., 2024) | 24.45 | 0.876 | 0.072 | 26.22 | 0.886 |
| **LearnIR (Ours)** | **28.87** | **0.970** | **0.049** | **27.93** | **0.921** |

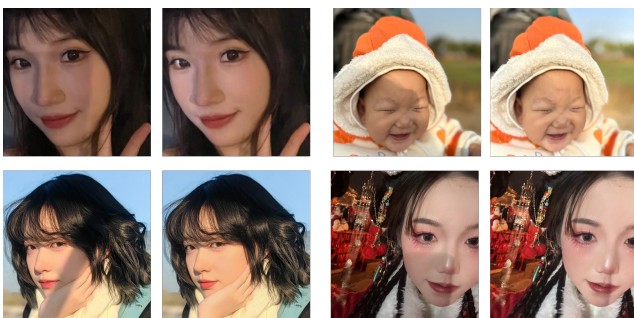

Figure 8: **Real face shadow data** Samples of real-world facial images with shadows, collected from the Internet. These images feature diverse identities and scene conditions, demonstrating realistic and challenging shadow patterns.

### A.5.5 LIMITATIONS

Despite LearnIR's strong performance, it faces certain limitations. As shown in Fig. 9, our model struggles to handle shadow regions with strong color casts (e.g., yellow or red-tinted shadows), often leading to residual artifacts that degrade visual quality. This highlights a challenge in generalization under complex lighting conditions and points to a promising future direction toward improving the robustness and adaptability of the proposed method in diverse real-world scenarios. Furthermore, the restoration quality is limited by its reliance on accurately learned gradient correction aligned with the true posterior.

### A.5.6 EVALUATION ON FACESHADOW DATASET

For completeness, quantitative comparisons in terms of PSNR, SSIM, and MAE on the FaceShadow datasets are presented in Table 5 due to space limitations in the main text.

### A.5.7 EXTENDED EVALUATION ON ADDITIONAL TEST DATA

**Real-World Results.** To evaluate the robustness of our face shadow removal model, we additionally collected a set of real-world facial images with shadows from the Internet. As illustrated in Fig. 8, our model effectively removes facial shadows, demonstrating its practical applicability and generalization ability in real-world conditions. However, as shown in Fig. 9, some challenging cases still remain.

### A.6 ADDITIONAL ABLATION STUDIES

### A.6.1 EFFECT OF DIFFUSION BACKBONES

As shown in Table 6, we compare three different diffusion backbones integrated with the same LearnIR (U-Net) module to evaluate their effectiveness in the residual prediction task. The results demonstrate that RDDM achieves the highest performance, with a PSNR of **28.87** and an SSIM of **0.970**, significantly outperforming DDIM and DiT-B/8. This suggests that the residual-based design

**Input**  **Output**  **GT**

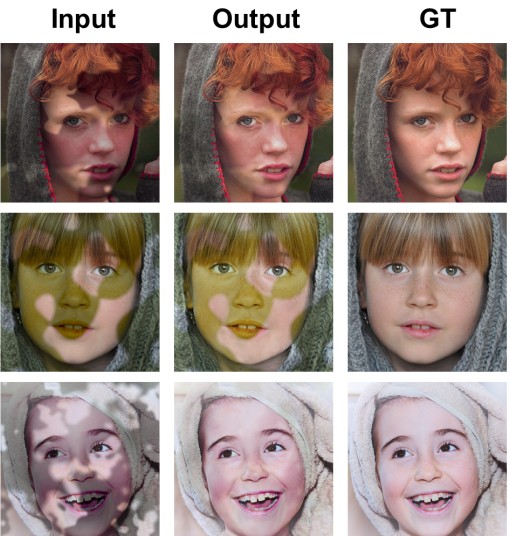

Figure 9: **(Failure Cases)** When shadow regions exhibit strong color tints (e.g., yellow or green filters), the model's shadow removal ability degrades significantly, resulting in noticeable residual artifacts that compromise visual quality.

Table 6: Quantitative comparison of different diffusion models with LearnIR (U-Net).

| Model | PSNR↑ | SSIM↑ |
|---|---|---|
| DiT-B/8 + LearnIR (U-Net) | 23.23 | 0.858 |
| DDIM + LearnIR (U-Net) | 26.11 | 0.887 |
| **RDDM + LearnIR (U-Net)** | **28.87** | **0.970** |

of RDDM facilitates more accurate noise estimation and better alignment with the target distribution. In contrast, DiT-B/8, which leverages a transformer-based structure, shows relatively lower performance, likely due to its weaker spatial inductive bias in pixel-level generation tasks. These findings validate the choice of RDDM as a more suitable backbone for residual-based diffusion frameworks.

A.6.2   ABLATION STUDY ON THE VAE AND DYNAMIC RESOLUTION MODULES

We conduct experiments on the FaceShadow dataset to evaluate the contributions of the Dynamic Resolution Module (DRM) and different image encoding strategies. Specifically, we fully fine-tune the 8× downsampling VAE from Stable Diffusion on all 30,000 face images. Our experiments demonstrate the clear advantages of the proposed Dynamic Resolution Module (DRM), which, combined with the DPSC backbone, forms our **LearnIR** approach. LearnIR significantly surpasses both the VAE baseline and the fixed-resolution setting in terms of performance and efficiency during the inference process, as detailed in Table 7.

Table 7: Impact of different image encoding strategies on model performance and efficiency. For the VAE variants, the reported PSNR and SSIM are shown as fine-tuned / frozen. "DPSC + VAE" denotes the 8× downsampling VAE from Stable Diffusion fully fine-tuned on all 30,000 FaceShadow images, while "freeze VAE" refers to the same VAE with parameters kept frozen during training.

| Variant | PSNR↑ | SSIM↑ | Params (M) ↓ | Inference Time (s) ↓ |
|---|---|---|---|---|
| **DRM (LearnIR)** | **28.78** | **0.970** | **139** | **0.276** |
| VAE(ft) / VAE(fr) | 26.63 / 26.85 | 0.901 / 0.951 | 475 | 0.532 |
| Fixed-Size | 27.12 | 0.956 | **139** | 1.253 |

Table 8: **Inference-time computational cost for LearnIR components.** We report the parameters, MACs, inference time, and GPU memory for the RDDM backbone, the DPSC gradient correction module, and the DRM dynamic-resolution interpolation. Notably, DRM uses non-learnable bilinear interpolation, resulting in MACs and memory so small that they are marked as "–" in the table.

| Module | Params ↓ | MACs (G) × Steps ↓ | Inference Time (s) ↓ | GPU Memory (GB) ↓ |
|---|---|---|---|---|
| RDDM | 516.5M | $1730.7 \times 5 = 8653.5$ | $0.217 \times 5 = 1.085$ | 9.8 GB |
| DRM | – | – | $8.23 \times 10^{-8} \times 5 = 4.15 \times 10^{-7}$ | – |
| DPSC | 139.4M | $233.6 \times 5 = 1168.0$ | $0.106 \times 5 = 0.531$ | 2.4 GB |
| **Full Model (LearnIR)** | **655.9M** | **$1964.3 \times 5 = 9821.5$** | **$0.323 \times 5 = 1.616$** | **12.2 GB** |

As shown in Table 7, integrating the Dynamic Resolution Module (DRM) with the DPSC backbone (**LearnIR**) yields clear benefits in both reconstruction quality and inference efficiency: LearnIR achieves 28.78 dB PSNR and 0.970 SSIM while reducing inference time to 0.276 s.

Notably, we observe that fully fine-tuning the 8× Stable Diffusion VAE on the 30,000 FaceShadow images does not improve the DPSC baseline in our setting. Specifically, the DPSC + VAE (fully fine-tuned) variant attains 26.63 dB / 0.901 SSIM, whereas the DPSC + VAE (frozen) counterpart achieves 26.85 dB / 0.951 SSIM. This indicates that full fine-tuning—without additional regularization or careful hyperparameter tuning—can perturb the pretrained latent space and hurt reconstruction quality (particularly SSIM).

Despite this, LearnIR (DPSC + DRM) still outperforms both VAE variants and the fixed-resolution baseline, demonstrating that the DRM effectively compensates for encoding-level differences and yields the best trade-off between quality and efficiency.

Table 8 provides an overview of the inference-time computational costs for the components of the LearnIR method, including the RDDM backbone, the DPSC gradient correction module, and the DRM dynamic-resolution interpolation. Notably, the DRM uses non-learnable bilinear interpolation, resulting in such minimal MACs and memory requirements that they are marked as "–" in the table. This table aids in understanding the resource consumption of each module during inference and the overall resource requirements of the LearnIR model.

## A.7 FORWARD PROCESS OF THE SDE

We introduce the variable transformation in Eq. 26:

$$\boldsymbol{x}_t = \tilde{\boldsymbol{x}}_t + \boldsymbol{R} \Rightarrow \tilde{\boldsymbol{x}}_t = \boldsymbol{x}_t - \boldsymbol{R} \tag{26}$$

Taking the differential on both sides of Eq. 26 yields:

$$d\boldsymbol{x}_t = d\tilde{\boldsymbol{x}}_t \tag{27}$$

Substituting Eq. 27 into the forward process of SDE gives:

$$d\tilde{\boldsymbol{x}} = -\frac{\beta(t)}{2}(\tilde{\boldsymbol{x}} + \boldsymbol{R})\, dt + \sqrt{\beta(t)}\, d\boldsymbol{w} \tag{28}$$

Reverting back to the original variable via $\boldsymbol{x}_t = \tilde{\boldsymbol{x}}_t + \boldsymbol{R}$, the forward SDE with residual can be expressed as:

$$d\boldsymbol{x} = -\frac{\beta(t)}{2}(\boldsymbol{x} - \boldsymbol{R})\, dt + \sqrt{\beta(t)}\, d\boldsymbol{w} \tag{29}$$

We can rewrite Eq. 29 into a general form:

$$d\boldsymbol{x} = f_t(\boldsymbol{x})\, dt + g_t\, d\boldsymbol{w}_t \tag{30}$$

where $f_t(\boldsymbol{x}) = -\frac{\beta(t)}{2}(\boldsymbol{x} - \boldsymbol{R})$ and $g_t = \sqrt{\beta(t)}$. This represents the standard form of a forward stochastic differential equation.

To facilitate the derivation of the reverse process, we interpret the continuous differential process in Eq. 30 as the limit of its discrete approximation as $\Delta t \to 0$:

$$\boldsymbol{x}_{t+\Delta t} = \boldsymbol{x}_t + f_t(\boldsymbol{x}_t)\,\Delta t + g_t\sqrt{\Delta t}\,\boldsymbol{\epsilon}, \quad \boldsymbol{\epsilon} \sim \mathcal{N}(0, \mathbf{I}) \tag{31}$$

## A.8 REVERSE PROCESS OF THE SDE

We reformulate the discrete approximation in Eq. 31 as a conditional probability:

$$p(\boldsymbol{x}_{t+\Delta t} \mid \boldsymbol{x}_t) = \mathcal{N}\left(\boldsymbol{x}_{t+\Delta t};\; \boldsymbol{x}_t + f_t(\boldsymbol{x}_t)\Delta t,\; g_t^2 \Delta t\,\mathbf{I}\right)$$
$$\propto \exp\left(-\frac{\|\boldsymbol{x}_{t+\Delta t} - \boldsymbol{x}_t - f_t(\boldsymbol{x}_t)\Delta t\|_2^2}{2g_t^2\Delta t}\right) \tag{32}$$

Eq. 32 describes the conditional probability distribution of the forward stochastic differential equation (SDE). Based on this, we use Bayes' rule to derive the conditional probability distribution of the reverse SDE, $p(\boldsymbol{x}_t \mid \boldsymbol{x}_{t+\Delta t})$:

$$p(\boldsymbol{x}_t \mid \boldsymbol{x}_{t+\Delta t}) \propto \frac{p(\boldsymbol{x}_{t+\Delta t} \mid \boldsymbol{x}_t)\,p(\boldsymbol{x}_t)}{p(\boldsymbol{x}_{t+\Delta t})}$$
$$\propto p(\boldsymbol{x}_{t+\Delta t} \mid \boldsymbol{x}_t) \exp\left(\log p(\boldsymbol{x}_t) - \log p(\boldsymbol{x}_{t+\Delta t})\right)$$
$$\propto \exp\left(-\frac{\|\boldsymbol{x}_{t+\Delta t} - \boldsymbol{x}_t - f_t(\boldsymbol{x}_t)\Delta t\|_2^2}{2g_t^2\Delta t} + \log p(\boldsymbol{x}_t) - \log p(\boldsymbol{x}_{t+\Delta t})\right) \tag{33}$$

As shown in Eq. 33, when $\Delta t$ is sufficiently small, the conditional probability $p(\boldsymbol{x}_{t+\Delta t} \mid \boldsymbol{x}_t)$ becomes significantly non-zero only when $\boldsymbol{x}_t$ is close to $\boldsymbol{x}_{t+\Delta t}$, and vice versa. Therefore, we can approximate the reverse process by considering only the local behavior where $\boldsymbol{x}_t$ and $\boldsymbol{x}_{t+\Delta t}$ are sufficiently close, for which a Taylor expansion is applicable.

$$\log p(\boldsymbol{x}_{t+\Delta t}) \approx \log p(\boldsymbol{x}_t) + (\boldsymbol{x}_{t+\Delta t} - \boldsymbol{x}_t) \cdot \nabla_{\boldsymbol{x}_t} \log p(\boldsymbol{x}_t) + \Delta t\,\frac{\partial}{\partial t} \log p(\boldsymbol{x}_t) \tag{34}$$

It is important to emphasize that the term $\frac{\partial}{\partial t} \log p(\boldsymbol{x}_t)$ cannot be neglected. This is because the probability density function $p(\boldsymbol{x}_t)$ describes the likelihood of the random variable taking the value $\boldsymbol{x}_t$ at time $t$, which inherently depends on both the temporal variable $t$ and the spatial variable $\boldsymbol{x}_t$. The quantity $p(\boldsymbol{x}_{t+\Delta t})$, in contrast, corresponds to the probability density at time $t + \Delta t$. Thus, when performing a Taylor expansion of $p(\boldsymbol{x}_t)$, the partial derivative with respect to time must be explicitly considered. Substituting Eq. 34 into Eq. 33 yields:

$$p(\boldsymbol{x}_t \mid \boldsymbol{x}_{t+\Delta t}) \propto \exp\left(-\frac{\left\|\boldsymbol{x}_{t+\Delta t} - \boldsymbol{x}_t - \left[f_t(\boldsymbol{x}_t) - g_t^2 \nabla_{\boldsymbol{x}_t} \log p(\boldsymbol{x}_t)\right]\Delta t\right\|^2}{2g_t^2\Delta t} + O(\Delta t)\right) \tag{35}$$

As $\Delta t \to 0$, the higher-order term $O(\Delta t)$ becomes negligible and can be omitted.

$$p(\boldsymbol{x}_t \mid \boldsymbol{x}_{t+\Delta t}) \propto \exp\left(-\frac{\left\|\boldsymbol{x}_{t+\Delta t} - \boldsymbol{x}_t - \left[f_t(\boldsymbol{x}_t) - g_t^2 \nabla_{\boldsymbol{x}_t} \log p(\boldsymbol{x}_t)\right]\Delta t\right\|^2}{2g_t^2\Delta t}\right)$$
$$\approx \exp\left(-\frac{\left\|\boldsymbol{x}_t - \boldsymbol{x}_{t+\Delta t} + \left[f_{t+\Delta t}(\boldsymbol{x}_{t+\Delta t}) - g_{t+\Delta t}^2 \nabla_{\boldsymbol{x}_{t+\Delta t}} \log p(\boldsymbol{x}_{t+\Delta t})\right]\Delta t\right\|^2}{2g_{t+\Delta t}^2\Delta t}\right) \tag{36}$$

Therefore, in the limit as $\Delta t \to 0$, the conditional distribution $p(\boldsymbol{x}_t \mid \boldsymbol{x}_{t+\Delta t})$ approximates a Gaussian with mean

$$\boldsymbol{x}_{t+\Delta t} - \left[ f_{t+\Delta t}(\boldsymbol{x}_{t+\Delta t}) - g_{t+\Delta t}^2 \nabla_{\boldsymbol{x}_{t+\Delta t}} \log p(\boldsymbol{x}_{t+\Delta t}) \right] \Delta t$$

and covariance $g_{t+\Delta t}^2 \Delta t \, \mathbf{I}$. We can thus rewrite this as Eq. 37:

$$\boldsymbol{x}_t = \boldsymbol{x}_{t+\Delta t} - \left[ f_{t+\Delta t}(\boldsymbol{x}_{t+\Delta t}) - g_{t+\Delta t}^2 \nabla_{\boldsymbol{x}_{t+\Delta t}} \log p(\boldsymbol{x}_{t+\Delta t}) \right] \Delta t + g_{t+\Delta t} \sqrt{\Delta t} \boldsymbol{\epsilon} \qquad (37)$$

We can further rewrite Eq. 37 in differential form to obtain the reverse-time SDE:

$$d\boldsymbol{x} = \left[ f_t(\boldsymbol{x}) - g_t^2 \nabla_{\boldsymbol{x}} \log p_t(\boldsymbol{x}) \right] dt + g_t \, d\boldsymbol{w} \qquad (38)$$

Substituting $f_t(\boldsymbol{x}) = -\frac{\beta(t)}{2}(\boldsymbol{x} - \boldsymbol{R})$ and $g_t = \sqrt{\beta(t)}$, we arrive at the final expression for the reverse-time SDE:

$$d\boldsymbol{x} = \left[ -\frac{\beta(t)}{2}(\boldsymbol{x} - \boldsymbol{R}) - \beta(t) \nabla_{\boldsymbol{x}_t} \log p_t(\boldsymbol{x}_t) \right] dt + \sqrt{\beta(t)} \, d\tilde{\boldsymbol{w}} \qquad (39)$$

## A.9 DERIVATION OF THE OBJECTIVE FUNCTION IN SCORE MATCHING MODELS

We start from the reverse-time stochastic differential equation (SDE), as shown in Eq. 39, and discretize it to perform iterative generation, leading to the following form:

$$\boldsymbol{x}_t - \boldsymbol{x}_{t+\Delta t} = - \left[ f_{t+\Delta t}(\boldsymbol{x}_{t+\Delta t}) - g_{t+\Delta t}^2 \nabla_{\boldsymbol{x}_{t+\Delta t}} \log p(\boldsymbol{x}_{t+\Delta t}) \right] \Delta t - g_{t+\Delta t} \sqrt{\Delta t} \cdot \boldsymbol{\epsilon}, \quad \boldsymbol{\epsilon} \sim \mathcal{N}(0, \mathbf{I}) \qquad (40)$$

which forms the basis of the generative process in diffusion models.

The main challenge lies in obtaining the term $\nabla_{\boldsymbol{x}} \log p_t(\boldsymbol{x})$, i.e., the gradient of the marginal distribution at time $t$. To this end, we utilize the analytical form of $p(\boldsymbol{x}_t \mid \boldsymbol{x}_0)$ to estimate $p(\boldsymbol{x}_t)$ indirectly, leading to the following identity:

$$\nabla_{\boldsymbol{x}_t} \log p(\boldsymbol{x}_t) = \mathbb{E}_{\boldsymbol{x}_0 \sim \tilde{p}(\boldsymbol{x}_0)} \left[ \nabla_{\boldsymbol{x}_t} \log p(\boldsymbol{x}_t \mid \boldsymbol{x}_0) \cdot w(\boldsymbol{x}_0) \right], \qquad (41)$$

where $w(\boldsymbol{x}_0)$ is a weight proportional to $p(\boldsymbol{x}_t \mid \boldsymbol{x}_0)$.

To avoid the computational cost of taking expectations over all training samples $\boldsymbol{x}_0$, we introduce a neural network $s_\theta(\boldsymbol{x}_t, t)$ to approximate the above gradient. Accordingly, we define the following least-squares loss:

$$\mathcal{L}(\theta) = \mathbb{E}_{\boldsymbol{x}_0, \boldsymbol{x}_t \sim p(\boldsymbol{x}_t \mid \boldsymbol{x}_0)} \left[ \| s_\theta(\boldsymbol{x}_t, t) - \nabla_{\boldsymbol{x}_t} \log p(\boldsymbol{x}_t \mid \boldsymbol{x}_0) \|^2 \right]. \qquad (42)$$

Minimizing this loss ensures that the network output converges to the true reverse-time gradient of the diffusion process, enabling effective modeling of the generative path.

## A.10 DETAILED DERIVATION OF SOLVING INVERSE PROBLEMS IN DIFFUSION MODELS

In the reverse-time process of stochastic differential equations (SDEs), we can introduce an inverse problem of the following form:

$$d\boldsymbol{x}_t = -\frac{\beta(t)}{2} \left( \boldsymbol{x}_t - \boldsymbol{R} \right) dt - \beta(t) \nabla_{\boldsymbol{x}_t} \log p_t(\boldsymbol{x}_t \mid \boldsymbol{y}) \, dt + \sqrt{\beta(t)} \, d\boldsymbol{w}_t, \qquad (43)$$

where $\boldsymbol{R}$ denotes the residual term and $\boldsymbol{w}_t$ is a standard Wiener process.

According to Bayes' rule, the posterior distribution can be obtained by combining the prior $p(\boldsymbol{x})$ and the observation likelihood $p(\boldsymbol{y} \mid \boldsymbol{x})$:

$$p(\boldsymbol{x} \mid \boldsymbol{y}) = \frac{p(\boldsymbol{y} \mid \boldsymbol{x}) \, p(\boldsymbol{x})}{p(\boldsymbol{y})}, \qquad (44)$$

which allows for inference of the unknown variable $\boldsymbol{x}$ conditioned on the observation $\boldsymbol{y}$. Based on this, the gradient term in Eq. 43 can be rewritten as:

$$\nabla_{\boldsymbol{x}_t} \log p_t(\boldsymbol{x}_t \mid \boldsymbol{y}) = \nabla_{\boldsymbol{x}_t} \log p_t(\boldsymbol{x}_t) + \nabla_{\boldsymbol{x}_t} \log p_t(\boldsymbol{y} \mid \boldsymbol{x}_t). \tag{45}$$

Substituting Eq. 45 into Eq. 43 yields:

$$\begin{aligned} d\boldsymbol{x}_t = &-\frac{\beta(t)}{2}\big(\boldsymbol{x}_t - \boldsymbol{R}\big)\,dt - \beta(t)\Big(\nabla_{\boldsymbol{x}_t} \log p_t(\boldsymbol{x}_t) + \nabla_{\boldsymbol{x}_t} \log p_t(\boldsymbol{y} \mid \boldsymbol{x}_t)\Big)dt \\ &+ \sqrt{\beta(t)}\,d\boldsymbol{w}_t. \end{aligned} \tag{46}$$

This decomposition explicitly separates the prior score $\nabla_{\boldsymbol{x}_t} \log p_t(\boldsymbol{x}_t)$ from the data-consistency score $\nabla_{\boldsymbol{x}_t} \log p_t(\boldsymbol{y} \mid \boldsymbol{x}_t)$, which is the basis of diffusion posterior sampling.

## A.11 DETAILED DERIVATION OF DIFFUSION POSTERIOR SAMPLING CORRECTION

We begin by writing the forward update in the residual-guided diffusion process as

$$\boldsymbol{z}_{t-1}^{\text{forward}} = \sqrt{\bar{\alpha}_{t-1}}\,\boldsymbol{z}_0^{(t)} + \big(1 - \sqrt{\bar{\alpha}_{t-1}}\big)\,\boldsymbol{R}_z + \sqrt{1 - \bar{\alpha}_{t-1}}\,\boldsymbol{\epsilon}, \tag{47}$$

where $\boldsymbol{R}_z = \boldsymbol{z}_y^{(t)} - \boldsymbol{z}_0^{(t)}$ and $\boldsymbol{\epsilon} \sim \mathcal{N}(\boldsymbol{0}, \mathbf{I})$.

On the other hand, the reverse prediction from the residual model is

$$\boldsymbol{z}_{t-1}^{\text{pred}} = \frac{1}{\sqrt{\alpha_t}}\,\boldsymbol{z}_t - \frac{1 - \alpha_t}{\sqrt{\alpha_t(1 - \bar{\alpha}_t)}}\,\boldsymbol{\epsilon}_\theta^{\text{res}}(\boldsymbol{z}_t, \boldsymbol{z}_y^{(t)}, t) + \sqrt{\frac{(1 - \bar{\alpha}_{t-1})(1 - \alpha_t)}{1 - \bar{\alpha}_t}}\,\boldsymbol{\epsilon}, \tag{48}$$

which corresponds to the parameterization in Definition 2.

To compare these two formulations, we express $\boldsymbol{z}_0^{(t)}$ with the standard reparameterization of the forward process (cf. Eq. 12) and substitute it into Eq. 47. After straightforward algebra, we obtain an explicit expression of the offset

$$\begin{aligned} \boldsymbol{z}_{t-1}^{\text{pred}} - \boldsymbol{z}_{t-1}^{\text{forward}} = &\left(\frac{1}{\sqrt{\alpha_t}} - \frac{2\sqrt{\bar{\alpha}_{t-1}} - 1}{\sqrt{\bar{\alpha}_t}}\right)\boldsymbol{z}_t \\ &- \big(1 - \sqrt{\bar{\alpha}_{t-1}}\big)\boldsymbol{z}_y^{(t)} \\ &- \frac{1 - \alpha_t}{\sqrt{\alpha_t(1 - \bar{\alpha}_t)}}\,\boldsymbol{\epsilon}_\theta^{\text{res}}(\boldsymbol{z}_t, \boldsymbol{z}_y^{(t)}, t) \\ &+ \left(\sqrt{\frac{(1 - \bar{\alpha}_{t-1})(1 - \alpha_t)}{1 - \bar{\alpha}_t}} - \sqrt{1 - \bar{\alpha}_{t-1}} + \frac{(2\sqrt{\bar{\alpha}_{t-1}} - 1)(1 - \alpha_t)}{\sqrt{\bar{\alpha}_t(1 - \bar{\alpha}_t)}}\right)\boldsymbol{\epsilon}. \end{aligned} \tag{49}$$

Since $\boldsymbol{\epsilon}$ is Gaussian and all the other terms are deterministic functions of $(\boldsymbol{z}_t, \boldsymbol{z}_y^{(t)}, t)$, the offset follows a Gaussian distribution

$$\boldsymbol{z}_{t-1}^{\text{pred}} - \boldsymbol{z}_{t-1}^{\text{forward}} \sim \mathcal{N}(\boldsymbol{\mu}, \sigma^2 \mathbf{I}), \tag{50}$$

where the mean and variance are given by

$$\boldsymbol{\mu}(\boldsymbol{z}_t, \boldsymbol{z}_y^{(t)}, t) = \left(\frac{1}{\sqrt{\alpha_t}} - \frac{2\sqrt{\bar{\alpha}_{t-1}} - 1}{\sqrt{\bar{\alpha}_t}}\right)\boldsymbol{z}_t - \big(1 - \sqrt{\bar{\alpha}_{t-1}}\big)\boldsymbol{z}_y^{(t)} - \frac{1 - \alpha_t}{\sqrt{\alpha_t(1 - \bar{\alpha}_t)}}\,\boldsymbol{\epsilon}_\theta^{\text{res}}(\boldsymbol{z}_t, \boldsymbol{z}_y^{(t)}, t), \tag{51}$$

$$\sigma^2(\boldsymbol{z}_t, \boldsymbol{z}_y^{(t)}, t) = \left(\sqrt{\frac{(1 - \bar{\alpha}_{t-1})(1 - \alpha_t)}{1 - \bar{\alpha}_t}} - \sqrt{1 - \bar{\alpha}_{t-1}} + \frac{(2\sqrt{\bar{\alpha}_{t-1}} - 1)(1 - \alpha_t)}{\sqrt{\bar{\alpha}_t(1 - \bar{\alpha}_t)}}\right)^2. \tag{52}$$

This shows that the offset between forward and predicted updates can be modeled as a Gaussian correction term, which forms the basis of our diffusion posterior sampling correction.

