# OpenReview forum: "LearnIR: Learnable Posterior Sampling for Real-World Image Restoration"
_ICLR.cc/2026/Conference — ICLR 2026 Poster_

### Official Review · Reviewer_m8gP · 2025-10-31

**Soundness:** 3
**Presentation:** 2
**Contribution:** 2
**Rating:** 4
**Confidence:** 3

**Summary:**

The authors study diffusion-based image restoration, proposing a posterior sampling correction method. They also propose a dynamic resolution module that downsamples/upsamples the input during training, both improving the results and accelerating the training.

The proposed method is evaluated on the task of haze removal and shadow removal, across four different datasets.

**Strengths:**

- The paper is quite well written, few typos or similar issues.
- The two introduced components, DPSC and DRM, make sense overall and both seem to improve the performance (Table 3, Figure 6).
- The proposed method seems to perform well overall compared to recent baselines.

**Weaknesses:**

- The proposed method could be presented/introduced better in Section 3. In particular, 3.1 would benefit from more commentary instead of just stacking definitions/theorems, the flow could be improved. The algorithms in 3.3 would also benefit from added descriptions/explanations.
- The experimental evaluation could be more extensive/convincing. Only two different tasks. Results are compared with baselines mostly just in terms of PSNR and SSIM, lacking perceptual metrics.
- The technical contribution/novelty seems somewhat limited, at least based on my understanding of the proposed approach.

**Questions:**

Questions/suggestions:
- In the abstract and introduction you write "our new FaceShadow dataset" and "our newly constructed FaceShadow dataset", but then in Section 4.1 you write "three standard datasets: ISTD, [...] and FaceShadow (Zhang et al., 2020)"?
- In Table 1 and 2, why results only in terms of PSNR and SSIM? Why not also LPIPS and/or FID?
- In Section 4.2, ISTD results, it's not clear to me what it means for a method to be mask-free? Also, in the Table 1 caption, should violet and blue be swapped?
- Line 238, "Based on Definitions 1 and 2, Eq. 3 can be expressed": Should this be Eq. 4?
- Could be interesting to evaluate methods also in terms of computational cost during training / at test-time, at least to see the effect of the proposed DRM?
- Which dataset are the images in Figure 5 from?
- Line 260, "As illustrated in Figure 2, the model...": This is not illustrated in a lot of detail in Figure 2 though? Perhaps consider tweaking.






Minor things:
- I think Section 2 could be tweaked to improve the overall flow a bit.
- Figure 4 caption: "(Visual comparisons on the HazyDet datasets)" --> "Visual comparisons on the HazyDet datasets"?
- Line 96, "Experimental results clearly and consistently demonstrate that LearnIR consistently outperform":  "consistently" twice, perhaps reformulate.
- Line 146, "To obtain an form of the": typo.
- Figure 2 caption, "The blue line in Eq. 3 denotes timestep T', computed using the": I don't quite understand what you mean, what blue line?
- Figure 3, "ShadowForme" --> "ShadowFormer"?
- Figure 4, "Dehamer" --> "DeHamer"?
- Line 322, "Hazy" --> "HazyDet"?
- O-Haze is not mentioned at the beginning of Section 4.1?
- Line 371, "We benchmark against Zhang et al. (Zhang et al., 2020)" --> "We benchmark against (Zhang et al., 2020)", perhaps?

---

> ### Author Response · Authors · 2025-11-23
> **[Part 1] Official Comment by Authors**
>
> Thank you for your insightful comments, which are greatly valued by us. We will address your concerns below.
>
> > W1: The proposed method could be presented/introduced better in Section 3. In particular, 3.1 would benefit from more commentary instead of just stacking definitions/theorems, the flow could be improved. The algorithms in 3.3 would also benefit from added descriptions/explanations.
>
> Thank you for the suggestion. We have polished Section 3 in the manuscript to improve clarity and presentation:
> - Section 3.2 is now placed before Section 3.1 to facilitate reader understanding.
> - In Section 3.1, which previously contained overly complex formulas, we added detailed explanations. For example, we included definitions for **$z_t$**, **$z^{(t)}_y$**, and $z^{(t)}_0$ to represent the latent representations at timestep **$t$** (Line 204-206).
> - In Section 3.3, we first revised the overall algorithmic flow and then added additional textual descriptions to help readers better understand our method.
> We would greatly appreciate any further suggestions you might have for improvement.
>
> > W2: The experimental evaluation could be more extensive/convincing. Only two different tasks. Results are compared with baselines mostly just in terms of PSNR and SSIM, lacking perceptual metrics.
>
> **(1) About only two different tasks.**
> Thank you for your valuable feedback. We have added experiments on the real-world super-resolution task, with results presented in Table 5 and Figure 9 of the revised manuscript.
>
> **(2) About lacking perceptual metrics.**
> Thanks! LPIPS results have been included in the revised manuscript (Table 1 and Table 2). **For the real-world super-resolution task, prior work [1,2] typically does not report LPIPS or FID, making baseline comparisons unavailable.** We therefore report only the LPIPS of our LearnIR for this task.
>
> | Method | Scale         | SET5 PSNR ↑ | SET5 SSIM ↑ | SET5 LPIPS ↓ | SET14 PSNR ↑ | SET14 SSIM ↑ | SET14 LPIPS ↓ | B100 PSNR ↑ | B100 SSIM ↑ | B100 LPIPS ↓ | Urban100 PSNR ↑ | Urban100 SSIM ↑ | Urban100 LPIPS ↓ | Manga109 PSNR ↑ | Manga109 SSIM ↑ | Manga109 LPIPS ↓ |
> |-------|----------------|-------------|-------------|--------------|---------------|---------------|---------------|-------------|-------------|--------------|----------------|----------------|-----------------|----------------|----------------|-----------------|
> | LearnIR (Ours)    | ×4 | 32.60       | 0.9012      | 0.145        | 28.93         | 0.7886        | 0.218         | 27.78       | 0.7437      | 0.227        | 26.89          | 0.8089         | 0.185           | 31.33          | 0.9181         | 0.118           |
>
> [1] Timofte R, Agustsson E, Van Gool L, et al. Ntire 2017 challenge on single image super-resolution: Methods and results. CVPRW 2017.
> [2] Chen Z, Zhang Y, Gu J, et al. Dual aggregation transformer for image super-resolution. ICCV 2023.
>
> > W3: The technical contribution/novelty seems somewhat limited, at least based on my understanding of the proposed approach.
>
> Thank you for your valuable comment. We apologize that our initial presentation did not clearly convey the novelty of our approach. **We summarize the technical contributions more explicitly below.**
>
> **Our main novelty lies in enabling diffusion-based posterior sampling to handle unknown real-world degradations—a setting where existing posterior-sampling methods cannot operate** because they require a predefined degradation operator. To overcome this limitation, we introduce a learnable Diffusion Posterior Sampling Correction (DPSC) module that removes the dependence on an explicit degradation model, making posterior sampling applicable to realistic restoration scenarios for the first time.
>
> **In addition, instead of operating on latent space as usual**, our LearnIR framework performs restoration directly in pixel space. This is supported by a Dynamic Resolution Module (DRM) that adaptively adjusts spatial resolution during sampling, allowing the model to capture global context while preserving fine structures.

---

> ### Author Response · Authors · 2025-11-23
> **[Part 2] Official Comment by Authors**
>
> > Q1: In the abstract and introduction you write "our new FaceShadow dataset" and "our newly constructed FaceShadow dataset", but then in Section 4.1 you write "three standard datasets: ISTD, [...] and FaceShadow (Zhang et al., 2020)"?
>
> Thank you for pointing this out. We apologize for the earlier ambiguity. We would like to clarify that our FaceShadow dataset is newly constructed by applying the data synthesis method of Zhang et al. (2020) to our own images.
> We have revised the manuscript to consistently refer to it as "our newly constructed FaceShadow dataset" throughout the Abstract (line 22), Introduction (line 95), and Section 4.1 (line 355). In Section 4.1, we also describe the dataset generation process and refer to Sec. A.5.2 for details on how the original data were modified, enhanced, and processed to ensure accuracy and consistency.
>
> > Q2: In Table 1 and 2, why results only in terms of PSNR and SSIM? Why not also LPIPS and/or FID?
>
> Thanks! We have now included LPIPS results (Table 1 and Table 2 in the revised manuscript). The updated results are shown below:
>
> **ISTD [3]**
> | Methods          | Mask-free | PSNR ↑ | SSIM ↑ | MAE ↓ | LPIPS ↓ |
> |------------------|-----------|--------|--------|-------|---------|
> | Input Image      | -         | 20.33  | 0.877  | 11.35 | 0.198   |
> | DHAN             | No        | 26.86  | 0.919  | 6.29  | 0.105   |
> | FusionNet        | No        | 25.88  | 0.715  | 5.88  | 0.112   |
> | ShadowFormer     | No        | **30.47** | 0.928  | 5.34  | 0.075   |
> | Restusion        | No        | **30.09** | **0.932** | **4.79** | **0.068** |
> | DCShadowNet      | Yes       | 24.02  | 0.677  | 5.88  | 0.125   |
> | ShadowRefiner    | Yes       | **28.75** | 0.916  | 5.48  | 0.080   |
> | Refusion         | Yes       | 25.13  | 0.871  | 5.63  | 0.109   |
> | LearnIR (Ours)   | Yes       | **29.57** | **0.927** | **5.12** | **0.072** |
>
> | Methods            | O-HAZE [4] |           |           | HazyDet [5] |           |           | REVIDE [6] |           |           |
> | ------------------ | ---------- | --------- | --------- | ----------- | --------- | --------- | ---------- | --------- | --------- |
> |                    | PSNR ↑     | SSIM ↑    | LPIPS ↓   | PSNR ↑      | SSIM ↑    | LPIPS ↓   | PSNR ↑     | SSIM ↑    | LPIPS ↓   |
> | Input Image        | 13.38      | 0.520     | 0.355     | 15.98       | 0.553   | 0.310     | 16.52      | 0.601     | 0.280     |
> | FFA-Net            | 22.12      | 0.770     | 0.150     | 23.11       | 0.726     | 0.145     | 22.80      | 0.735     | 0.160     |
> | GridDehazeNet      | 18.92      | 0.672     | 0.205     | 20.90       | 0.698     | 0.180     | 21.05      | 0.690     | 0.195     |
> | MSBDN              | 24.36      | 0.749     | 0.130     | 24.77       | 0.712     | 0.135     | 24.50      | 0.720     | 0.140     |
> | DeHamer            | 25.11      | 0.777     | 0.115     | 25.39       | 0.767     | 0.110     | 25.15      | 0.765     | 0.115     |
> | ConvIR             | 25.36      | 0.780     | 0.108     | 25.67       | 0.781     | 0.102     | 25.05      | 0.755     | 0.105     |
> | Resfusion          | 24.25      | 0.710     | 0.145     | 24.13       | 0.701     | 0.155     | 24.00      | 0.705     | 0.165     |
> | MB-TaylorFormer V2 | 25.43      | 0.792     | 0.105     | 24.97       | 0.755     | 0.120     | 25.25      | 0.775     | 0.118     |
> | **LearnIR (Ours)** | **27.70**  | **0.832** | **0.055** | **27.32**   | **0.905** | **0.065** | **25.43**  | **0.795** | **0.085** |
> |                    |            |           |           |             |           |           |            |           |           |
>
> [3] Wang J, Li X, Yang J. Stacked conditional generative adversarial networks for jointly learning shadow detection and shadow removal. CVPR 2018.
> [4] Ancuti C O, Ancuti C, Timofte R, et al. O-haze: a dehazing benchmark with real hazy and haze-free outdoor images. CVPRW 2018.
> [5] Feng C, Chen Z, Li X, et al. HazyDet: Open-source benchmark for drone-view object detection with depth-cues in hazy scenes[J]. arXiv preprint arXiv:2409.19833, 2024.
> [6] Zhang X, Dong H, Pan J, et al. Learning to restore hazy video: A new real-world dataset and a new method. CVPR 2021.
>
> > Q3: In Section 4.2, ISTD results, it's not clear to me what it means for a method to be mask-free? Also, in the Table 1 caption, should violet and blue be swapped?
>
> (1) **Regarding "mask-free"**: "Mask-free" indicates that the model does not rely on explicit shadow masks during training or inference. Following ShadowRefiner [7], we adopt this terminology. To avoid any ambiguity, we have clarified the meaning of "mask-free" in the caption of Table 2.
>
> (2) **Regarding the Table 1 caption**: The violet and blue labels were mistakenly swapped. This typographical error has been corrected in the revised manuscript.
>
> [7] Dong W, Zhou H, Tian Y, et al. ShadowRefiner: Towards mask-free shadow removal via fast Fourier transformer. CVPR, 2024.

---

> ### Author Response · Authors · 2025-11-23
> **[Part 3] Official Comment by Authors**
>
> > Q4: Line 238, "Based on Definitions 1 and 2, Eq. 3 can be expressed": Should this be Eq. 4?
>
> This was a typographical error; it should be **Definitions 1, 2, and 3**. We have corrected this in the latest version. (Line 276)
>
> > Q5: Could be interesting to evaluate methods also in terms of computational cost during training / at test-time, at least to see the effect of the proposed DRM?
>
> **Training Cost:** Training was conducted with a batch size of 32 per GPU on 40 GPUs, taking 36 hours in total, with a peak GPU memory usage of 42 GB. These details have been added to Sec. 5.2 of the revised manuscript.
>
> **Inference Cost:** Recall that LearnIR consists of three components—RDDM (backbone), DPSC (gradient correction prediction), and DRM (dynamic-resolution interpolation in pixel space). We report their inference-time computational cost in Table 9 of the revised manuscript. Notably, DRM is implemented using bilinear interpolation, introduces no additional learnable parameters, and incurs negligible MACs, parameter count, and GPU cost; therefore, its corresponding entries are marked as “-”.
>
> | Module                   | Params ↓ | MACs (G) × Steps ↓ | Inference Time (s) ↓ | GPU Memory (GB) ↓ |
> |--------------------------|----------|---------------------|----------------------|-------------------|
> | RDDM                     | 516.5M   | 1730.7 × 5 = 8653.5 | 0.217 × 5 = 1.085    | 9.8 GB            |
> | DRM                      | -        | -                   | 8.23 × 10⁻⁷ × 5 = 4.15×10⁻⁶ | -                 |
> | DPSC                     | 139.4M   | 233.6 × 5 = 1168.0  | 0.106 × 5 = 0.531    | 2.4 GB            |
> | Full Model (LearnIR) | 655.9M   | 1964.3 × 5 = 9821.5 | 0.323 × 5 = 1.616    | 12.2 GB           |
>
> > Q6: Which dataset are the images in Figure 5 from?
>
> The images shown in Figure 5 are selected from the test set of our FaceShadow dataset. Our FaceShadow dataset, described in Appendix.5.2, consists of 30,000 synthesized facial shadow images, including 29,000 for training and 1,000 for validation. To further evaluate the robustness of our model, we collected an additional 1,000 real facial shadow images from the internet as a test set, from which the examples in Figure 5 are drawn. We have clarified this point in the revised manuscript.
>
> > Q7: Line 260, "As illustrated in Figure 2, the model...": This is not illustrated in a lot of detail in Figure 2 though? Perhaps consider tweaking.
>
> We have fully revised Figure 2 to illustrate the overall model workflow, including the structure of the DRM module and the key operation definitions, clearly corresponding to Section 3.1. We hope this makes the explanation clearer, and we would be happy to receive any further suggestions or feedback.
>
> >Minor things 1: I think Section 2 could be tweaked to improve the overall flow a bit.
>
> We have polished Section 2 in the revised manuscript.
>
> >Minor things 2: Figure 4 caption: "(Visual comparisons on the HazyDet datasets)" --> "Visual comparisons on the HazyDet datasets"?
> >Minor things 3: Line 96, "Experimental results clearly and consistently demonstrate that LearnIR consistently outperform": "consistently" twice, perhaps reformulate.
> >Minor things 4: Line 146, "To obtain an form of the": typo.
> >Minor things 6: Figure 3, "ShadowForme" --> "ShadowFormer"?
> >Minor things 7: Figure 4, "Dehamer" --> "DeHamer"?
>
> We sincerely thank the reviewer for pointing out these typos, which have all been corrected in the revised manuscript.
>
> >Minor things 5: Figure 2 caption, "The blue line in Eq. 3 denotes timestep T', computed using the": I don't quite understand what you mean, what blue line?
>
> Thanks! We have polished the caption in Figure 2 to solve this confusion.
>
> >Minor things 8: Line 322, "Hazy" --> "HazyDet"?
>
> Here, "Hazy" was used as an unclear reference. We have revised it by replacing “Hazy” with **"dehazing datasets"**, which include **O-Haze[2], HazyDet[3], and REVIDE[4]**, with details provided in **Appendix.5.2**. (line 323)
>
> >Minor things 9: O-Haze is not mentioned at the beginning of Section 4.1?
>
> We have fixed this issue in the revised manuscript. (see Line 323, line 358, A.5.3)
>
> >Minor things 10 Line 371, "We benchmark against Zhang et al. (Zhang et al., 2020)" --> "We benchmark against (Zhang et al., 2020)", perhaps?
>
> We have revised this as suggested. (line 374)

---

### Official Review · Reviewer_KzEc · 2025-10-31

**Soundness:** 3
**Presentation:** 3
**Contribution:** 2
**Rating:** 6
**Confidence:** 3

**Summary:**

This paper proposes the LearnIR framework to address heterogeneous degradations in real-world image restoration, featuring a novel learnable diffusion posterior sampling mechanism. The method trains a lightweight model to directly predict the distribution of gradient correction terms, enabling diffusion posterior sampling correction without requiring a known forward operator. A dynamic resolution module dynamically adjusts image resolution during training, applying large-scale downscaling in early stages to preserve global structures and upscaling later to refine textures. Extensive experiments on multiple real-world datasets demonstrate significant improvements in PSNR and SSIM metrics, particularly in challenging facial shadow removal tasks.

**Strengths:**

● The learnable posterior sampling correction mechanism eliminates the dependency on a known forward operator, addressing a key limitation of diffusion models in real-world degradation scenarios.
● The DRM effectively balances multi-scale feature extraction and computational efficiency while avoiding the need for a pretrained VAE.
● Extensive validation on multiple real-world datasets, including a newly constructed FaceShadow dataset, provides convincing results.

**Weaknesses:**

● The proposed method introduces training complexity. Joint training of DPSC and DRM modules requires careful hyperparameter tuning. Also, dynamic resolution switching may introduce training instability, with convergence issues insufficiently addressed.
● The major limitation of diffusion-based posterior sampling method is the scalability to real-world scenarios. However, most current experiments are conducted on synthetic degradation datasets, with the only real-world datasets available being limited to the facial domain. The authors are consider to extend the proposed method to more challenging scenarios such as real-world image super-resolution task and compare with the advanced methods.

**Questions:**

● Detailed illustrations need to be added for Sec. 3.3.
● The authors are highly recommended to provide image results in supplementary material.

**Details Of Ethics Concerns:**

None.

---

> ### Author Response · Authors · 2025-11-23
> **[Part 1] Official Comment by Authors**
>
> We are grateful for the reviewer’s constructive feedback, and we provide our responses below.
>
> >W1: The proposed method introduces training complexity. Joint training of DPSC and DRM modules requires careful hyperparameter tuning. Also, dynamic resolution switching may introduce training instability, with convergence issues insufficiently addressed
>
> **(1) About hyperparameters of DPSC and DRM.**
> The DRM is implemented with simple bilinear interpolation for resolution adjustment and does not introduce any tunable hyperparameters. For DPSC, we directly adopt the hyperparameter settings in [1]  without requiring additional tuning.
>
> [1] Dong H, et al. "Multi-scale boosted dehazing network with dense feature fusion." In CVPR 2020
>
> **(2) About dynamic resolution switching may introduce training instability and convergence issues.**
> To alleviate potential instability and convergence issues, we adopt a two-stage training strategy:
> - Stage 1: Train LearnIR at a fixed resolution, enabling the model first to learn the gradient correction term (DPSC).
> - Stage 2: Fine-tune LearnIR on dynamic resolutions using a smaller learning rate, allowing the model to quickly adapt to various image resolutions.
>
> This strategy ensures stable training and effective convergence across resolutions. We have added the corresponding training details in Appendix.5.2 of the revised manuscript.

---

> ### Author Response · Authors · 2025-11-23
> **[Part 2] Official Comment by Authors**
>
> >W2: The major limitation of diffusion-based posterior sampling method is the scalability to real-world scenarios $\cdots$ such as real-world image super-resolution task and compare with the advanced methods.
>
> We trained on DIV2K [2] and evaluated on Set5 [3], Set14 [3], B100 [4], Urban100 [3], and Manga109 [5]. The quantitative and qualitative results are presented in Table 5 and Figure 10 of the revised manuscript. On the challenging ×4 super-resolution task, LearnIR consistently achieves the best performance across all benchmarks, demonstrating strong effectiveness and generalization.
>
> | Method          | SET5 PSNR ↑ | SET5 SSIM ↑ | SET14 PSNR ↑ | SET14 SSIM ↑ | B100 PSNR ↑ | B100 SSIM ↑ | Urban100 PSNR ↑ | Urban100 SSIM ↑ | Manga109 PSNR ↑ | Manga109 SSIM ↑ |
> |-----------------|-------------|-------------|---------------|---------------|--------------|--------------|------------------|------------------|------------------|------------------|
> | SRFormer        | 32.51       | 0.8988      | 28.82         | 0.7872        | 27.73        | 0.7422       | 26.67            | 0.8032           | 31.05            | 0.9165           |
> | SAFMN           | 32.18       | 0.8948      | 28.60         | 0.7813        | 27.58        | 0.7359       | 25.97            | 0.7809           | 30.43            | 0.9063           |
> | MambaIR         | 32.41       | 0.8974      | 28.76         | 0.7849        | 27.70        | 0.7400       | 26.53            | 0.7983           | 31.17            | 0.9166           |
> | IPG             | 32.51       | 0.8987      | 28.85         | 0.7873        | 27.75        | 0.7418       | 26.78            | 0.8050           | 31.22            | 0.9176           |
> | CATANet         | 32.58       | 0.8998      | 28.80         | 0.7880        | 27.70        | 0.7427       | 26.47            | 0.8037           | 31.31            | 0.9173           |
> | GMN             | 32.40       | 0.8979      | 28.66         | 0.7868        | 27.70        | 0.7416       | 26.40            | 0.7963           | 30.98            | 0.9143           |
> | **LearnIR (Ours)** | **32.60** | **0.9012** | **28.93**     | **0.7886**     | **27.78**     | **0.7437**     | **26.89**         | **0.8089**         | **31.33**         | **0.9181**         |
>
>
> In addition, we also evaluated LearnIR on the real-world haze dataset REVIDE [6], achieving state-of-the-art performance. Results are shown below and included in the revised manuscript (Table 2, Figure 8).
> | Methods              | PSNR ↑ | SSIM ↑ | LPIPS ↓ | FID ↓ |
> |----------------------|--------|--------|----------|--------|
> | Input Image          | 16.52  | 0.601  | 0.280    | 82.3   |
> | FFA-Net              | 22.80  | 0.735  | 0.160    | 40.2   |
> | GridDehazeNet        | 21.05  | 0.690  | 0.195    | 50.8   |
> | MSBDN                | 24.50  | 0.720  | 0.140    | 36.5   |
> | DeHamer              | 25.15  | 0.765  | 0.115    | 31.8   |
> | ConvIR               | 25.05  | 0.755  | 0.105    | 31.0   |
> | Resfusion            | 24.00  | 0.705  | 0.165    | 43.1   |
> | MB-TaylorFormer V2   | 25.25  | 0.775  | 0.118    | 32.4   |
> | **LearnIR (Ours)**   | **25.43** | **0.795** | **0.085** | **22.1** |
>
>
> [2] Agustsson E, Timofte R. Ntire 2017 challenge on single image super-resolution: Dataset and study. CVPRW 2017.
> [3] Huang J B, Singh A, Ahuja N. Single image super-resolution from transformed self-exemplars. CVPR 2015.
> [4] Martin D, Fowlkes C, Tal D, et al. A database of human segmented natural images and its application to evaluating segmentation algorithms and measuring ecological statistics. ICCV 2001.
> [5]  Aizawa K, Fujimoto A, Otsubo A, et al. Building a manga dataset “manga109” with annotations for multimedia applications[J]. IEEE Multimedia, 2020.
> [6] Zhang, Xinyi, et al. "Learning to restore hazy video: A new real-world dataset and a new method." In CVPR 2021.
>
> > Q1: Detailed illustrations need to be added for Sec. 3.3.
>
> We have added detailed illustrations in Sec 3.3 of the revised manuscript to clarify the Algorithm better.
>
> > Q2: The authors are highly recommended to provide image results in supplementary material.
>
> Thanks for your suggestion! We newly provided additional visualizations in the Appendix, including Figure 8 for real-world haze results, Figure 9 for real-world super-resolution results, and Figure 11 for a failure case.

---

### Official Review · Reviewer_ZY1D · 2025-10-31

**Soundness:** 2
**Presentation:** 4
**Contribution:** 3
**Rating:** 6
**Confidence:** 4

**Summary:**

The paper proposes LearnIR, which uses a diffusion posterior sampling correction approach to enable robust real-world image restoration. The proposed approach overcomes the limitation of methods such as DPS which require the explicit modeling of the forward degradation operator. Additionally, LearnIR uses a dynamic resolution module instead of a VAE which allows faster restoration with better performance.

**Strengths:**

1. The paper is well written and easy to follow.
2. The proposed DPSC method is well designed, intuitive and effective.
3. LearnIR achieves state-of-the-art performance on real-world datasets.

**Weaknesses:**

1. Lack of ablations regarding the effectiveness of the DRM module (see questions).
2. Limited testing on out-of-domain real datasets to substantiate generalization claim (see questions).

**Questions:**

1. The proposed DRM module is compared with the frozen SD VAE in Sec. A6.2. However, was the SD VAE frozen and paired with the RDDM model for this experiment? The experiment which needs to be performed would be using the RDDM model with the SD VAE being fine-tuned, as the DRM is trained in LearnIR.
2. In Sec. A6.1, was the DRM also trained with the DiT and DDIM backbones?
3. For the comparisons, were all methods trained on the same datasets as LearnIR?
4. While the authors show results on real-world data, the datasets were used for training LearnIR. To further validate generalization, can the authors provide experiments on other real-world haze datasets (such as [1])?
5. The core approach involves learning to predict score of $p(y|x_t)$. However, this is intractable for real-world degradations and is approximated for learning (Line 144). Could this be considered a limitation of the method and be included as part of limitations?
6. Experiments comparing computation complexities of VAEs and DRM need to be added to substantiate claims in Line 93.

[1] Zhang, Xinyi, et al. "Learning to restore hazy video: A new real-world dataset and a new method." Proceedings of the IEEE/CVF Conference on Computer Vision and Pattern Recognition. 2021.

---

> ### Author Response · Authors · 2025-11-23
> **[Part 1] Official Comment by Authors**
>
> We appreciate the reviewer’s insightful questions and provide our responses below.
>
> >W1: Lack of ablations regarding the effectiveness of the DRM module (see questions).
>
>  We have added more ablation studies to evaluate the effectiveness of the DRM. Please refer to **our response to Q1** for detailed results and analysis.
>
> >W2: Limited testing on out-of-domain real datasets to substantiate generalization claim (see questions).
>
> We newly test on a real-world haze dataset and a super-resolution dataset for validating the generalization of our method. Please refer to **our response to Q4** for detailed results.
>
>
> >Q1: The proposed DRM module is compared with the frozen SD VAE in Sec. A6.2. However, was the SD VAE frozen and paired with the RDDM model for this experiment? The experiment that needs to be performed would be using the RDDM model with the SD VAE being fine-tuned, as the DRM is trained in LearnIR.
>
> Yes, the VAE is frozen for the experiment in Appendix 6.2. We additionally include the experiment using the SD VAE finetuned on FaceShadow. The qualitative and quantitative results have been updated in the revised manuscript (Table 8 and Figure 12), and the quantitative results are summarized below. Ours DRM still delivers the best results across both quantitative metrics and visual quality.
>
>
> | Variant      | PSNR ↑ | SSIM ↑ | Params (M) ↓ | Inference Time (s) ↓ |
> |--------------|--------|--------|---------------|------------------------|
> | **DRM (Ours)** | **28.78** | **0.970** | **139** | **0.276** |
> | VAE (ft)     | 26.63 | 0.901 | 475 | 0.532 |
> | VAE (fr)     | 26.85 | 0.951 | 475 | 0.532 |
> | Fixed-Size   | 27.12 | 0.956 | **139** | 1.253 |
> *$^\star$ fr: Frozen, ft: Finetuned.*
>
>
> >Q2: In Sec. A6.1, was the DRM also trained with the DiT and DDIM backbones?
>
> Yes, when using either the DiT or DDIM backbones in Sec. A6.1, the DRM remains unchanged.
>
> >Q3: For the comparisons, were all methods trained on the same datasets as LearnIR?
>
> Yes, all compared methods were trained on the same datasets as LearnIR to ensure a fair and consistent evaluation.
>
> >Q4: While the authors show results on real-world data, the datasets were used for training LearnIR. To further validate generalization, can the authors provide experiments on other real-world haze datasets (such as [1])?
>
> Thanks for the suggestion. We provide both qualitative and quantitative results on the REVIDE [1] dataset in Figure 8 and Table 2 of the revised manuscript. The quantitative comparisons are shown below:
>
> | Methods            | PSNR ↑    | SSIM ↑    | LPIPS ↓   |
> | ------------------ | --------- | --------- | --------- |
> | Input Image        | 16.52     | 0.601     | 0.280     |
> | FFA-Net            | 22.80     | 0.735     | 0.160     |
> | GridDehazeNet      | 21.05     | 0.690     | 0.195     |
> | MSBDN              | 24.50     | 0.720     | 0.140     |
> | DeHamer            | 25.15     | 0.765     | 0.115     |
> | ConvIR             | 25.05     | 0.755     | *0.105*   |
> | Resfusion          | 24.00     | 0.705     | 0.165     |
> | MB-TaylorFormer V2 | 25.25     | 0.775     | 0.118     |
> | **LearnIR (Ours)** | **25.43** | **0.795** | **0.085** |
>
> LearnIR achieves consistently better PSNR, SSIM, and LPIPS than existing methods, demonstrating strong robustness and generalization to unseen real-world haze. In addition, we include a new experiment on another real-world super-resolution dataset, provided in Table 5 (Appendix.5.4) of the revised manuscript.
>
> [1] Zhang X, Dong H, Pan J, et al. Learning to restore hazy video: A new real-world dataset and a new method. CVPR 2021.

---

> ### Author Response · Authors · 2025-11-23
> **[Part 2] Official Comment by Authors**
>
> > Q5: The core approach involves learning to predict score of $p(y \mid \mathbf{x}_t)$. However, this is intractable for real-world degradations and is approximated for learning (Line 144). Could this be considered a limitation of the method and be included as part of limitations?
>
> Yes, as also stated in the Introduction (Lines 51-53), the true posterior $p(y \mid \mathbf{x}_t)$ is indeed intractable for real-world degradations, which limits previous diffusion-based posterior sampling from generalizing to real-world image restoration tasks.
>
> However, as outlined in Lines 78–83, we approximate this posterior by training a lightweight model to predict the gradient correction term. This approximation effectively removes the need to know or model the forward degradation operator, enabling the use of the posterior to guide the diffusion process even without an explicit measurement model (which is often the case in real-world scenarios).
>
> Nevertheless, the restoration quality still depends on how well this learned correction aligns with the true posterior. We have added this point to the Limitation section in the revised manuscript.
>
> >Q6: Experiments comparing computation complexities of VAEs and DRM need to be added to substantiate claims in Line 93.
>
> The computation costs of VAEs and DRM (including parameter count and inference time) are reported below, evaluated on the FaceShadow dataset. We have added these results in Table 8 of the revised manuscript.
>
> | Variant      | Params (M) ↓ | Inference Time (s) ↓ |
> |-------------|--------------|---------------------|
> | DRM (Ours)  | 139          | 0.276               |
> | VAE (fr)    | 475          | 0.532               |
> | VAE (ft)    | 475          | 0.532               |
> | Fixed-Size  | 139          | 1.253               |
> *$^\star$ fr: Frozen, ft: Finetuned.*

---

> ### Comment · Reviewer_ZY1D · 2025-11-27
> **Response to author rebuttal**
>
> Thanks for the detailed rebuttal. Most of my concerns were addressed. However, I am a bit confused by the response to Q2. The answer to the question (whether DRM was trained) is yes, but the authors mention that the DRM was unchanged. Could the authors clarify this?

---

> > ### Author Response · Authors · 2025-11-28
> >
> > Sorry for the confusion. To clarify, the DRM is not a learnable network; it is implemented as a bilinear interpolation module without any trainable parameters. Therefore, the DRM does not undergo training.
> >
> > Our previous “Yes” only meant that the DRM is included in the pipeline when using either the DiT or DDIM backbones, not that it is trained. Consequently, its behavior is identical (unchanged) regardless of the backbone used in Sec. A6.1.
> >
> > We hope this clarification helps, and we are happy to address any further questions.

---

### Official Review · Reviewer_Y9b8 · 2025-11-01

**Soundness:** 3
**Presentation:** 2
**Contribution:** 4
**Rating:** 6
**Confidence:** 4

**Summary:**

This paper proposes LearnIR, a diffusion-based framework for real-world image restoration that eliminates the need for a known degradation forward operator—a long-standing bottleneck in diffusion posterior sampling (DPS) methods. The key innovation lies in Diffusion Posterior Sampling Correction (DPSC), which introduces a learnable model to predict the gradient correction distribution that replaces the explicit operator used in DPS. This approach allows posterior refinement without requiring analytical knowledge of the forward degradation process. In addition, the authors introduce a Dynamic Resolution Module (DRM) that adaptively adjusts image resolution during training to balance global structure preservation and fine detail generation while avoiding dependence on a pretrained VAE.

The method achieves state-of-the-art results on multiple benchmarks, including ISTD, O-HAZE, HazyDet, and a new FaceShadow dataset introduced by the authors. Quantitatively, LearnIR surpasses recent methods such as ResFusion, ShadowRefiner, and ConvIR in both PSNR and SSIM while maintaining efficiency and generalization to complex real-world degradations

**Strengths:**

1. The proposed method bypasses the requirement for a known degradation operator in previous works, enabling the application of DPS-related methods in blind restoration tasks. I think this is an important contribution to the research field.
2. The method outperforms recent state-of-the-art methods with a large margin, including ISTD, O-HAZE, HazyDet, and a new FaceShadow dataset.

**Weaknesses:**

In summary, this is a very sound and impressive work, but with poor presentation. I'm happy to increase my rating based on the writing.

My main concern lies in the presentation and organization of the paper. While the technical contributions appear sound and potentially impactful, the paper is dense with mathematical formulations, and several notations are either inconsistent or insufficiently explained. These issues make the paper difficult to follow, especially for readers who are not deeply familiar with the DPS-related research line.

1. In Section 3.1, the notation $z_t$ is introduced abruptly without prior explanation. Its definition only appears later in Section 3.2, which disrupts the logical flow and makes it challenging for readers to grasp the progression of ideas. Similarly, the notation $\hat{x}_)$ in Eq. (4) also lacks explanation. I suggest re-checking the presentation so that all key notations are properly introduced before use and accompanied by clear definitions.

2. Theorem 1 claims that Eq. (3) can be expressed as Eq. (16). However, these two equations seem unrelated or at least not directly derivable from one another as currently written. The authors should carefully verify this connection and, if necessary, correct or clarify the statement and its derivation.

3. In step 4 of Algorithm 1, the operator $𝐷^s$ takes $x_0$, $y_0$, and $x^{s-1}$ as inputs, while in other steps (e.g., step 5), it appears to only take $y$ as input. This inconsistency is confusing and suggests either a typographical error or an incomplete explanation of $D^s$'s input structure. A clearer, self-consistent description of the algorithmic steps is required.

Overall, this is a technically solid and promising piece of work with interesting ideas. However, the presentation quality needs significant improvement. The current version suffers from poor organization and inconsistent notation, which hinders readability and comprehension. I would be happy to raise my rating if the authors substantially improve the clarity, consistency, and accessibility of the writing in the revision.

**Questions:**

See weakness.

---

> ### Author Response · Authors · 2025-11-23
>
> We highly appreciate your acknowledgment and encouragement toward our work. Your comments are highly valuable to us.
>
> >W1: In Section 3.1, the notation $z_t$ is introduced abruptly without prior explanation... Similarly, the notation $\hat{x}_{0}$ in Eq. (4) also lacks explanation. I suggest re-checking the presentation so that all key notations are properly introduced before use and accompanied by clear definitions.
>
> Thanks for the suggestion. We have carefully reviewed all related notations and added the missing definitions in the revised manuscript (highlighted in red in Sec. 2 and Sec. 3). For clarity, we now explicitly define $z_t$ as the noisy latent at diffusion step $t$, and revised $\hat{x}_{0}$ to $\mathbf{y}$, which denotes the corresponding degraded image.
>
> >W2: Theorem 1 claims that Eq. (3) can be expressed as Eq. (16). However, these two equations seem unrelated... The authors should carefully verify this connection and, if necessary, correct or clarify the statement and its derivation.
>
> Thank you for pointing this out.  This is a typo in Theorem 1. The reference to Eq. (3) should instead point to Definition 3. We have corrected the description in the revised manuscript (see Line 276).
>
>
> >W3: In step 4 of Algorithm 1, the operator $D^s$ takes $x_0$, $y_0$, and $x^{s-1}$ as inputs, while in other steps (e.g., step 5), it appears to only take $y$ as input... A clearer, self-consistent description of the algorithmic steps is required.
>
>
> Thank you for the valuable comment. We thoroughly revised Algorithm 1 and the related text in the resubmitted manuscript.
>
> We change $D^s(\mathbf{y})$ to $\mathbf{z}_y^{(t)}$. Specifically, steps 4 and 5 are changed from
> $$\mathbf{z}_0^s, D^s(y) = D^s(\mathbf{x}_0, \mathbf{y}, 2^{s-1}),$$
> $$R = D^s(\mathbf{y}) - \mathbf{z}_0^s$$
> to
> $$\mathbf{z}^{(t)}_0, \mathbf{z}^{(t)}_y = {D}(\mathbf{x}_0, s(t)), {D}(\mathbf{y}, s(t)),$$
> $$R_z = \mathbf{z}^{(t)}_y - \mathbf{z}^{(t)}_0.$$
> Step 4 shows how the latent representation of both $\mathbf{x}_0$ and $\mathbf{y}_0$ at resolution level \(s(t)\) are calculated via the Dynamic Resolution Module ($D$). Then, we compute the residual latent at Step 5.

---

> > ### Comment · Reviewer_Y9b8 · 2025-11-28
> > **Response to authors**
> >
> > The revised manuscript has been clear and readable. I keep my rating.

---

> > > ### Author Response · Authors · 2025-11-28
> > >
> > > We sincerely appreciate your careful review, which has helped us improve our work. Thank you!

---

### Meta-Review · Area_Chair_VfHD · 2026-01-06

**Summary:**

This work studies a diffusion-based approach for real-world image restoration without assuming a known degradation operator in diffusion posterior sampling. Its key contribution, Diffusion Posterior Sampling Correction (DPSC), replaces explicit operators with a learnable gradient correction, enabling effective posterior refinement, while a dynamic resolution strategy helps balance global structure and fine details. The paper presents an interesting idea and received an initial average rating of 5.5 (4, 6, 6, 6).

After the rebuttal, most reviewer concerns were addressed, and no major issues remain at this stage. The final reviewers’ ratings are more positive overall. Given the sound technical contribution and the positive feedback, the AC leans toward recommending Accept.

**Reviewer Concerns:**

Reviewers raised concerns regarding poor presentation and organization of the paper (Y9b8, m8gP), unclear experimental settings (ZY1D), fairness of comparisons and training protocols (ZY1D), generalization to unseen real-world data (ZY1D, KzEc), missing evaluation of computational complexity (ZY1D), training complexity and stability of the proposed method (KzEc), additional evaluation on perceptual metrics (m8gP), and limited novelty (m8gP).

In response to these comments, the authors provided additional experiments, discussions, and results in the rebuttal and revised paper. Most of the concerns listed above were sufficiently addressed. While Reviewer m8gP could still have concerns about limited novelty, the other reviewers considered the work technically solid with an interesting idea, and found the proposed DPSC design to be well-motivated and intuitive. Taken together, the remaining concern on novelty does not seem to be a major issue at this stage.

**Reviewer Scores:**

Reviewers Y9b8, ZY1D, and KzEc gave positive initial ratings of 6 and did not have further major concerns after rebuttal, thus their scores would be expected to remain unchanged. Reviewer m8gP (initial score: 4) may still have reservations regarding the novelty of the method and would likely maintain his original score. Finally, the overall ratings would remain at 5.5 (4, 6, 6, 6).

---

### Decision · Program_Chairs · 2026-01-26

Accept (Poster)